# Report from the 30th Meeting on Toxinology, “Unlocking the Deep Secrets of Toxins”, Organized by the French Society of Toxinology on 2–3 December 2024

**DOI:** 10.3390/toxins17020094

**Published:** 2025-02-17

**Authors:** Pascale Marchot, Ziad Fajloun, Évelyne Benoit, Sylvie Diochot

**Affiliations:** 1CNRS/Aix-Marseille Université, Laboratoire Architecture et Fonction des Macromolécules Biologiques (AFMB), Faculté des Sciences—Campus Luminy, F-13288 Marseille Cedex 09, France; 2Laboratory of Applied Biotechnology (LBA3B), Department of Cell Culture, AZM Center for Research in Biotechnology and Its Applications, Doctoral School for Sciences and Technology, Lebanese University, Tripoli 1300, Lebanon; ziad.fajloun@ul.edu.lb; 3Department of Biology, Faculty of Sciences 3, Campus Michel Slayman Ras Maska, Lebanese University, Tripoli 1352, Lebanon; 4Service d’Ingénierie Moléculaire Pour la Santé (SIMoS), EMR CNRS/CEA 9004, Département Médicaments et Technologies Pour la Santé (DMTS), Institut des Sciences du Vivant Frédéric Joliot, Université Paris-Saclay, CEA, F-91191 Gif-sur-Yvette, France; evelyne.benoit@cea.fr; 5Institut de Pharmacologie Moléculaire et Cellulaire, Université Côte d’Azur, CNRS UMR7275, INSERM U1323, Sophia Antipolis, F-06560 Valbonne, France

**Keywords:** algal toxin, animal toxin, bacterial toxin, biological activity, environmental factor, fungal toxin, mold toxin, mycotoxin, phycotoxin, plant toxin, structure–function relationship, therapeutic drug, venomics

## Abstract

The French Society of Toxinology (SFET) held its 30th Annual Meeting (RT30) on 2–3 December 2024 at Hôtel Le Saint Paul in Nice, France, on the beautiful French Riviera. It was the first time that the event was organized outside of Paris. The meeting brought together 74 participants and focused on the main theme, “Unlocking the Deep Secrets of Toxins”, which delved into cutting-edge research in the field of animal venoms and toxins from animal, plant, fungal, algal, mold and bacterial sources. The event emphasized the dynamic and ever-evolving nature of toxins, often influenced by environmental factors, their interactions with molecular or cellular ligands, their mechanisms of action and their potential applications in therapy. These key topics were explored in depth during oral communications and poster sessions across three main thematic areas, each dedicated to a specific aspect of toxinology. A fourth, more general session provided an opportunity for participants to present recent work that fell outside the main themes but still contributed valuable insights to the broader field. This report presents the abstracts of seven of the invited lectures, fifteen of the selected lectures and sixteen of the posters, following the authors’ agreement to publish them. Additionally, the names of the “Best Oral Communication” and “Best Poster” awardees are highlighted, recognizing the outstanding contributions made by early-career researchers and their innovative work in toxinology.

## 1. Foreword

We warmly acknowledge the contributions of all those colleagues who work tirelessly to ensure the national and international prominence of the French Society of Toxinology (SFET) (http://sfet.asso.fr/international, English site accessed since 16 February 2010). We are also deeply grateful to everyone who helped make the 30th Meeting on Toxinology (RT30), held on 2–3 December 2024 at Hôtel Le Saint Paul in Nice, France, a resounding success. This RT30 meeting marked the first time this annual event was held outside Paris, this year on the French Riviera. Special thanks go to our dedicated long-standing sponsors and to those who joined us for the first time, for their invaluable support (Figure 1).

## 2. Scope and Topics of the SFET Meeting

The RT30 meeting attracted 74 attendees, comprising 42% women and 58% men. Of them, 25 participants came from European countries other than France, including Belgium, the Czech Republic, Denmark, England, Germany, Ireland, Italy, Monaco, Portugal and Spain, while 5 came from other continents, including Tunisia, Brazil and the USA. This large ratio of foreign participants highlights the international appeal of the RT30 event and clearly testifies to the continued relevance and attractiveness of the SFET meetings. This diverse gathering brought together students (26%), young and confirmed researchers (67%) and professionals (7%) from around the world, all eager to share knowledge, foster or consolidate collaborations and discuss the latest advancements in the field of toxinology. The substantial international presence also reflects the growing global interest in the scientific developments being presented and discussed during the meeting. As such, the SFET meetings continue to serve as a significant platform for advancing the field and prompting international collaboration in toxinology.

The central theme of this Toxinology Meeting, “Unlocking the Deep Secrets of Toxins”, was extensively explored through a diverse series of presentations by 22 distinguished speakers. These talks provided a comprehensive overview of the vast and varied types of toxins that are known to date, covering areas such as bacterial toxins, animal toxins, phycotoxins and fungal or mold mycotoxins. The presentations revealed that these toxins are in a continuous state of evolution, often influenced by environmental factors such as climate change or the complex predator/prey relationships observed in venomous animals. The speakers discussed the dynamic nature of toxin development, emphasizing that these toxins adapt to their surroundings in response to ecological pressures, creating new challenges for researchers. The discovery of these toxins is made even more complex by the need for cutting-edge technological tools. These innovations are not only crucial for identifying new toxins, but also enable detailed screenings of their biological activity, particularly their interactions with receptors or ion channels. This evolving field continues to push the boundaries of science, offering exciting new opportunities for both basic and applied toxinology research.

In particular, seven toxinology experts from England, Italy, Finland and France delivered insightful presentations on a range of key topics in the field. Their contributions included (i) a detailed exploration of the molecular interactions between pathogenic bacteria and host cells from a structural perspective, shedding light on the intricate mechanisms by which bacterial toxins influence cellular functions; (ii) significant advancements in the detection and monitoring of mycotoxins and phycotoxins, with a particular focus on how these toxins evolve in response to climate change and shifting environmental conditions; and (iii) the growing importance of studying animal toxins, not only to better understand the biology of venomous species, but also to harness these toxins for the development of novel diagnostic and therapeutic tools. These presentations emphasized the multidisciplinary nature of toxinology and its crucial role in addressing both basic scientific questions and practical challenges in medicine and environmental science.

Fifteen other speakers, chosen on abstract from a diverse group of researchers, post-doctoral fellows and students, also shared their valuable insights on various topics related to animal and bacterial toxins and to phyco- and mycotoxins. These presentations highlighted the latest advancements in the field, exploring new findings and the evolving understanding of how these toxins function and impact biological systems. Several of these presentations specifically focused on the molecular interactions between toxins and ion channels that have significant therapeutic potential. Others described processes for detecting and purifying toxins from environmental or animal sources, the latter being a potential source of cell proliferation inhibitors in cancer models. These talks emphasized the promising opportunities for drug development and diagnostic tools that could emerge from a deeper understanding of these interactions, which could lead to innovative treatments for a range of diseases and conditions.

Among the sixteen posters presented, eleven were also featured as 3 min flash talks. Some of these brief but impactful presentations highlighted cutting-edge methods for screening active animal toxins that interact with ion channels of therapeutic interest in diseases such as cardiomyopathies, muscle disorders, pain and neurological conditions. Others were more fundamental and exploratory in nature, focusing on the biochemical properties, mode of action or toxicity of peptidic toxins and proteins with enzymatic properties. These flash talks demonstrated the innovative approaches being used to explore how animal toxins could lead to the development of novel treatments or diagnostic tools for various medical conditions, emphasizing the potential of toxinology in advancing therapeutic strategies.

Thanks to a generous donation from the SFET, two awards of EUR 200 each were granted to young toxinologists: one for the best oral communication and one for the best poster (Figure 2). The awardees were selected through a voting process carried out by a distinguished committee comprising the invited speakers and members of the SFET Board of Directors present at the RT30 meeting. The selection process recognized outstanding contributions to the field, rewarding the presenters for their innovative research and impactful presentations, further encouraging excellence in toxinology.

Above all, we extend our heartfelt thanks to the Editors of *Toxins* for their support in publishing a Special Issue titled “Unlocking the Deep Secrets of Toxins.” This issue aims at featuring both the report from the RT30 meeting and a collection of peer-reviewed original articles and reviews on the theme of the RT30 meeting and beyond. We are confident that this Special Issue appeals to a broad audience, including those colleagues who were unable to attend the RT30 meeting. Additionally, it will serve as a comprehensive resource for researchers and students in the field of toxinology.

## 3. Scientific and Organizing Committee (SFET Board of Directors in 2024)

Évelyne Benoit, Commissariat à l’énergie atomique et aux énergies alternatives (CEA) de Saclay, Gif-sur-Yvette, FranceKatrina Campbell, Institute for Global Food Security, Queen’s University Belfast, Belfast, Northern IrelandAlexandre Chenal, Institut Pasteur, Paris, FranceSylvie Diochot, Institut de Pharmacologie Moléculaire et Cellulaire (IPMC), Valbonne, FranceSébastien Dutertre, Institut des Biomolécules Max Mousseron (IBMM), Montpellier, FranceZiad Fajloun, Doctoral School of Science and Technology, Lebanese University, Tripoli, LebanonDaniel Ladant, Institut Pasteur, Paris, FranceChristian Legros, Université d’Angers, Angers, FrancePascale Marchot, Centre National de la Recherche Scientifique (CNRS)/Aix-Marseille Université, Marseille, FranceMichel R. Popoff, retired from Institut Pasteur, Paris, FranceLoïc Quinton, Université de Liège, Liège, BelgiumMichel Ronjat, retired from l’institut du thorax, Université de Nantes, Nantes, France

## 4. Invited Lectures (When More than One Author, the Underlined Name Is That of the Presenter)

### 4.1. Climate and Mycotoxins, Deciphering the Dynamics of Change


**Marco Camardo Leggieri * and Paola Battilani**


Dipartimento di Scienze Delle Produzioni Vegetali Sostenibili, Università Cattolica del Sacro Cuore, Via Emilia Parmense, 84, 29122 Piacenza, Italy.

***** Correspondence: marco.camardoleggieri@unicatt.it

**Abstract:** Climate change is a complex phenomenon capable of modifying ecosystems globally. The increase in temperature, currently in line with the forecast of +2 °C, accompanied by the well-known phenomena of increase in the concentration of CO_2_ in the air, variation in the distribution and intensity of rainfall and the occurrence of extreme events, has already shown significant effects in different areas of the world. To be taken into serious consideration is the impact that climate change can have on food safety risks, an area in which mycotoxins play a fundamental role. Higher temperatures are favoring the migration, introduction and greater abundance of thermophilic and thermotolerant fungal species; consequently, some mycotoxigenic fungi, such as *Aspergillus flavus* and *Fusarium graminearum*, are extending their respective diffusion areas. The increase in temperature shortens the growth cycle of crops and allows changes in the respective cultivation areas. Furthermore, abiotic stressors resulting from climate change are expected to weaken the resistance of host crops, making them more vulnerable to fungal infections while also increasing the risk of mycotoxin contamination. Fungi–host-plant interactions are expected to vary with climate change, with changes also to the structure of microbial communities, influencing the prevalence and co-occurrence of mycotoxins in the future. Expanding research on different pathosystems, in addition to the much more studied cereals, also including the mycotoxins currently considered minor, and the possibility of known mycotoxigenic fungi to colonise new host crops, are all aspects of absolute importance. Future research efforts should focus on studying how the co-occurrence of mycotoxigenic fungi might change in a climate change scenario, as well as how climate change might interfere with the production of modified mycotoxins, resulting from cross-talk between the plant and the fungus, constantly influenced by the climate. Studying and understanding the evolving complexity is essential to guarantee food safety and protection and prevent harmful effects on human and animal health.

**Keywords:** climate change; crop; food safety; mycotoxin

### 4.2. Toxin Delivery: Architecture, Biogenesis and Mechanism of Action of a Bacterial Nanoweapon, the Type VI Secretion System


**Éric Cascales ***


Laboratoire d’Ingénierie des Systèmes Macromoléculaires (LISM), Institut de Microbiologie, Bioénergies et Biotechnologie (IM2B), CNRS—Aix-Marseille Université, Marseille, France.

***** Correspondence: cascales@imm.cnrs.fr

**Abstract:** Polymorphic toxins are modular proteins comprising an N-terminal cargo domain, which specifies their transport, fused to a domain carrying various toxic activities. Some of the cargo domains convey the toxin to a fascinating nanoweapon, the type VI secretion system (T6SS). The T6SS uses a contractile mechanism to deliver protein effectors to diverse cell types of prokaryotic and eukaryotic origins, therefore participating in inter-bacterial competition and pathogenesis. The T6SS comprises an envelope-spanning complex anchoring a cytoplasmic tubular edifice. This tubular structure is evolutionarily, functionally and structurally related to the tail of contractile phages. It is composed of an inner tube tipped by a spike complex and engulfed within a sheath-like structure. This structure assembles onto a platform called “baseplate” that is recruited and anchored to the membrane complex. The T6SS functions as a nanospeargun: upon sheath contraction, the inner tube is propelled into the target cell, enabling the delivery of highly potent toxins. I will talk about the architecture, biogenesis and mode of action of this fascinating secretion machine and will present the activity and mechanism of delivery of selected toxins.

**Keywords:** contractile mechanism; inter-bacterial competition; nanoweapon; protein effector; secretion system; toxin delivery; toxin loading

### 4.3. Discovery and Development of Small-Molecule Drugs for the Treatment of Snakebite Envenoming


**Nicholas R. Casewell ***


Centre for Snakebite Research & Interventions, Liverpool School of Tropical Medicine, Liverpool L3 5QA, UK

***** Correspondence: nicholas.casewell@lstmed.ac.uk

**Abstract:** Snakebite envenoming is a neglected tropical disease that causes >100,000 deaths annually, while as many as 400,000 survivors are said to be left with long-term morbidity each year as the result of a bite. Envenoming pathologies are diverse as the result of venom variation but can be categorized as being primarily related to hemotoxicity (bleeding, coagulopathy), neurotoxicity (paralysis, respiratory failure) and/or cytotoxicity (local tissue damage at the bite site). Polyclonal-antibody-based antivenoms are effective if given promptly after a bite, but this is challenging in the rural tropics because travel times to the clinical environment required for antivenom delivery are long, often resulting in poor patient outcomes. There is therefore a compelling need to develop new snakebite therapeutics that can be administered in a community setting soon after a bite. Repurposed small-molecule drugs that inhibit specific snake venom toxins show considerable promise for tackling this issue because their defined safety profiles and oral bioavailability are amenable for rapid use. In this talk, I will describe some of the recent progress in this space, with a focus on oral drugs that target the snake venom metalloproteinase toxins that cause hemorrhage, coagulopathy and local tissue damage following viper bites. The talk will focus on the preclinical and clinical development of such repurposed molecules, including the progression of 2,3-dimercapto-1-propanesulfonic acid (unithiol) into clinical trials. I will also highlight the potential value of drug combination therapies for more broadly treating diverse types of snakebite, as well as highlighting ongoing research to discover new drugs through medicinal chemistry approaches and plans for evaluating the efficacy of oral snake venom metalloproteinase inhibitors in future Phase II clinical trials.

**Keywords:** metalloproteinase; oral drug; snakebite

### 4.4. Harmful Algal Blooms and Biotoxins: A One Health Challenge


**Marie-Yasmine Dechraoui Bottein ***


ECOSEAS, CNRS, Université Côte d’Azur, Nice, France.

***** Correspondence: marie-yasmine.bottein@univ-cotedazur.fr

**Abstract:** Harmful algal blooms (HABs) pose a significant threat to our ecosystems, economy, food safety and public health. Certain algal species produce potent toxins that can accumulate in marine organisms and eventually enter the human food chain. Additionally, high-biomass HABs can lead to ocean oxygen depletion and massive fish kills. Events associated with HABs, particularly those involving benthic species, are linked to habitat changes, biodiversity loss and climate change, as well as to increasing coastal development and demand for seafood. This study explores the intersection of environmental, social, animal health and human health disciplines, emphasizing the perspectives of various stakeholders, including coastal communities and industries, on scales ranging from local to global. We highlight the necessity of an integrated, transdisciplinary, science-based approach to enhance the prediction of HAB events, improve early warning systems and develop effective responses. Ultimately, this approach aims to reduce the impacts of HABs while supporting sustainable development.

**Keywords:** biotoxin; early warning system; food safety; harmful algal bloom; One Health Challenge

### 4.5. Synthesis and Pathogenesis of Botulinum Neurotoxin


**Maria B. Nowakowska ^1^, Katja Selby ^1^, Adina Przykopanski ^2^, Maren Krüger ^3^, Linfeng Gao ^4^, Nigel P. Minton ^5^, François P. Douillard ^1^, Martin B. Dorner ^3^, Brigitte G. Dorner ^3^, Rongsheng Jin ^4^, Andreas Rummel ^2^, Miia Lindström ^1,^***


^1^ Department of Food Hygiene and Environmental Health, Faculty of Veterinary Medicine, University of Helsinki, Helsinki, Finland^2^ Institute for Toxicology, Hannover Medical School, Hannover, Germany^3^ Biological Toxins, Centre for Biological Threats and Special Pathogens, Robert Koch Institute, Berlin, Germany^4^ Department of Physiology and Biophysics, University of California Irvine, USA.^5^ Clostridia Research Group, BBSRC/EPSRC Synthetic Biology Research Centre (SBRC), Biodiscovery Institute, School of Life Sciences, University of Nottingham, Nottingham, UK

***** Correspondence: miia.lindstrom@helsinki.fi

**Abstract:** Botulinum neurotoxins (BoNTs) are produced by various species of clostridia and occasionally by other bacteria. BoNT is encoded within the neurotoxin gene cluster (NGC) along with neurotoxin-associated proteins (NAPs). All NGCs encode the non-toxic non-hemagglutinin (NTNH) protein, which forms a protective complex with BoNT. Whereas some NGCs encode three NAPs called hemagglutinins (HA33, HA70 and HA17), which are complexed with NTNH-BoNT to assist oral absorption, other NGCs encode instead four NAPs, namely OrfX1, OrfX2, OrfX3 and P47, with unknown biological functions. Here, we investigated the biological role of OrfX/P47 proteins co-produced with BoNT-NTNH in a mouse model. Using the CRISPR-Cas9 genome editing tool, we first generated a library of *Clostridium botulinum* Group II type E Beluga mutants with single or multiple gene deletions and/or nonsense mutations in *orfX/p47* genes. We validated and characterized these mutants using in vitro and in vivo assays and extracted BoNT-NTNH and NAPs from these mutants. Finally, we conducted mouse oral feeding experiments using these different cell extracts of BoNT, NTNH and OrfX/P47 proteins and demonstrated the contributing role of each OrfX/P47 protein in enhancing the oral toxicity of BoNT in mice. These findings provide new insights into the potential functions of OrfX/P47 in BoNT pathogenesis.

**Keywords:** *Clostridium botulinum*; neurotoxin; OrfX/P47 protein

### 4.6. The Yin and Yang of Venom and Mammalian sPLA2s: From Evolution to Revolution to Therapeutic Avenues


**Gérard Lambeau ***


Institute of Molecular and Cellular Pharmacology, CNRS, Inserm and Université Côte d’Azur, Valbonne Sophia Antipolis, France.

***** Correspondence: lambeau@ipmc.cnrs.fr

**Abstract:** Secreted phospholipases A2 (sPLA2s) are remarkable enzymes, among the smallest. They were discovered in venoms but are also present in mammals, plants, fungi, viruses and bacteria. sPLA2s are small proteins of 14–19 kDa sharing a conserved active site stabilized by many disulfides. They are interfacial enzymes, with a complex enzymatic activity toward phospholipids organized in aggregates. The enzyme binds to the lipid interface to then hydrolyze phospholipids, releasing fatty acids and lysophospholipids. Importantly, each sPLA2 has specific enzymatic properties toward cell membranes, extracellular vesicles, lipoproteins, dietary lipids, lung surfactants and membranes from pathogens. sPLA2s can produce potent lipid mediators and modify cellular lipid metabolism. Last but not least, some sPLA2s have natural active site mutations yet are toxic, and both active and inactive sPLA2s can bind to receptors and soluble proteins, bringing the concept that sPLA2s are enzymes and ligands. Importantly, the binding of sPLA2s to these proteins may promote or inhibit their action.

In this talk, I will discuss the molecular evolution and function of venom and mammalian sPLA2s, up to their potential as therapeutic targets in human diseases. First, I will highlight their molecular diversity in venoms and underscore how this diversity has led to the discovery of 12 mammalian sPLA2 genes and that of additional sPLA2s in snakes, beyond the venom gland. Second, I will provide examples of the biological functions of sPLA2s and their relationship to enzymatic activity. Third, I will pinpoint how the toxic and pharmacological activities of venom and mammalian sPLA2s can be controlled by inhibitors, natural or synthetic. Fourth, I will summarize the good, the bad and the ugly of the 30-year odyssey to develop small inhibitors of mammalian sPLA2s by pharmaceutical companies, which ended up failing in clinical trials for inflammatory diseases but also provided new hope for snakebite envenomation. Specific second-generation inhibitors are clearly needed. Finally, I will update our knowledge on PLA2R1, the best characterized sPLA2 receptor, which turned out to be the major autoantigen in membranous nephropathy, a rare but severe autoimmune kidney disease. This has led to a revolution in patients’ diagnosis and healthcare in the last 15 years. In the next decade, mammalian sPLA2s may become game changers in infectious diseases and beyond, without forgetting their “old” venom counterparts, as illustrated by the snake in the caduceus of physicians and pharmacists, originating from Asclepios and Hygeia in the Greek mythology, about 25 centuries ago.

**Keywords:** enzyme; human disease; inhibitor; lipid; molecular evolution; receptor; secreted phospholipase A2; snake venom

### 4.7. **Interferon: Tug of War Between Host and Pathogen**


**Charlotte Odendall ***


Sir Henry Dale Fellow, King’s College London, WC2R 2LS, Dept of Infectious Diseases, Guy’s Hospital, London SE1 9RT, UK.

***** Correspondence: charlotte.odendall@kcl.ac.uk

**Abstract:** Type I and III Interferons (IFNs) are part of the arsenal of immune mediators produced by innate immune cells upon detection of pathogens. This presentation will examine the interplay between enteric bacterial pathogens and IFNs. We will discuss how pathogenic inhibition of IFN signaling pathways by *Shigella* favors bacterial intracellular replication infection and colonization of the murine gut. This work reveals novel roles of these IFN families in bacterial infection, which will be further examined in the context of *Salmonella* pathogenesis.

**Keywords:** interferon; IFN-stimulated genes; *Salmonella*; T3SS

## 5. Oral Presentations (When More than One Author, the Underlined Name Is That of the Presenter)

### 5.1. Development of an Electrochemical Model for the Detection and Evaluation of the Potential Toxicity of Naturally Occurring Furanic Compounds


**Imène Ayaden ^1,^*, Céline Hoffmann ^2^, Chouaha Bouzidi ^1^, Thomas Gaslonde ^1^, Joëlle Perard ^1^, Florence Souquet ^1^, Xavier Cachet ^1^**


^1^ Cibles Thérapeutiques et Conception de Médicaments CiTCoM, UMR8038, CNRS, Faculté de Pharmacie de l’Université Paris Cité, Paris, France^2^ Unité de Technologies Chimiques et Biologiques Pour la Santé (UTCBS) Inserm U1267 CNRS UMR8258, Faculté de Pharmacie de l’Université Paris Cité, Paris, France

***** Correspondence: imeneayaden@gmail.com

**Abstract:** Environmental furanic compounds of plant and fungal origin represent a significant risk to human health in affecting vital organs, in particular liver and lung depending on the route of exposure. These compounds can be specialized metabolites produced by plants or microorganisms or neo-formed contaminants prevalent in heat-treated foods, industrial emissions and tobacco smoke. They undergo metabolic activation mediated by cytochrome P450 (CYP450) enzymes. While liver toxicity is well documented, emerging research highlights the critical role of CYP450 enzymes in lung tissues, specifically isoform CYP2E1, which is notably expressed in pulmonary epithelial cells. These enzymes are responsible for bioactivating inhaled furans, leading to the formation of reactive metabolites such as 2-cis-butene-1,4-dial. These intermediates interact with endogenous nucleophiles, resulting in oxidative stress, protein adducts and DNA damage that can trigger cellular injury and inflammation in the lungs. Chronic exposure to furanic compounds through inhalation has been associated with a range of lung pathologies, from acute bronchial injury and alveolar edema to chronic inflammation and fibrosis. Animal studies have demonstrated that inhaled furans selectively target club cells within the bronchioles due to their high expression of CYP2E1, leading to localized tissue damage. In severe cases, prolonged exposure may increase the risk of developing pulmonary neoplasms. The rising levels of environmental furans, driven by industrial pollution and global warming, further amplify the need for evaluating their toxicological impact. Therefore, there is a clear need for a mechanism-based approach to screen potentially toxic furans and establish systematic structure–toxicity relationships. Our research focuses on developing a simple electrochemical model that simulates the metabolic activation of furans by CYP450 enzymes. This approach aims to establish a link between the oxidation potential of furanic compounds and their potential toxicity. In addition, a bioactivation model based on human hepatoma HepaRG cells was developed to correlate the electrochemical model and evaluate the toxicity of furans. Inhibition of CYP450 could be used to assess the oxidation of furan rings, as studies have shown that furans can act as covalent inhibitors of CYP450 following bioactivation (“mechanism-based inhibition” (MBI)). We present herein the feasibility study conducted on model molecules that demonstrates the relevance of our model as a predictive tool for assessing the risks of exposure to furans in the environment.

**Keywords:** CYP450; bioactivation; electrochemical model; environmental toxin; furanic compound; structure–toxicity relationship

### 5.2. Venomous Voltage: Unraveling the Interactions Between Toxins and Ion Channels


**Anissa Bara ^1,^*, Kim Boddum ^1^, Line Ledsgaard ^2^, Aneesh Karatt-Vellatt ^3^, Melisa Bernard Valle ^2^, Anne Damsbo ^2^, Charlotte Rimbault ^2^, Damian Bell ^1^, Andreas H. Laustsen ^2^**


^1^ Sophion Bioscience, Ballerup, Denmark^2^ Department of Biotechnology & Biotherapeutics, Technical University of Denmark, Lyngby, Denmark^3^ IONTAS Ltd., Cambridgeshire, UK

***** Correspondence: aba@sophion.com

**Abstract:** Snakebite was designated Neglected Tropical Disease status by the World Health Organization (2017), causing 100,000 yearly deaths and around 400,000 amputations. Each snake species has a unique venom, often consisting of dozens of different toxins. The century-old, traditional technique to generate snake antivenoms involved purifying antibodies from horse blood serum following immunization with snake venom. However, there are several drawbacks: equine–human immunoreactivity and side effects, batch-to-batch variation and specificity to the snake venom used. In the last decade, advances in antibody engineering have made antibody discovery and development more efficient and specific, including creating human recombinant antivenom antibodies to target and neutralize key toxin peptides. One of the most medically relevant groups of snake toxins is the α-neurotoxins, targeting the muscle alpha1-nicotinic acetylcholine receptor (alpha1-nAChR). For over two decades, automated patch-clamp systems have been used to advance our understanding of ion channel biophysics, pharmacology and roles in physiology and disease. Here, using QPatch II & Qube 384 automated patch-clamp, we functionally evaluate snake venom α-neurotoxins and antivenom, toxin-neutralizing IgG monoclonal antibodies on the muscle alpha1-nAChR and in vivo efficiency.

**Keywords:** antivenom; electrophysiology; high-throughput

### 5.3. Algal Toxins and Seafood Safety in the Mediterranean Sea


**Guillaume Barnouin ^1,^*, James Kennedy ^1^, Nathalie Hilmi ^2^, Rachel Clausing ^3^, Antoine Lafitte ^4^, Marie-Yasmine Dechraoui Bottein ^1^**


^1^ Université Côte d’Azur, CNRS, ECOSEAS, Nice, France.^2^ Centre Scientifique de Monaco, Monaco.^3^ Dipartimento di Scienze della Terra, dell’Ambiente e della Vita, Università di Genova, Genova, Italy.^4^ UNEP/MAP, Plan Bleu, Marseille, France.

***** Correspondence: guillaume.barnouin@etu.univ-cotedazur.fr

**Abstract:** Harmful algal blooms are the naturally occurring proliferation of microscopic algae that have adverse effects on the marine environment as well as on animal and human health, constituting a One Health issue. Biotoxin-producing harmful algal blooms are increasingly prevalent in the Mediterranean Sea, posing significant threats to marine life and humans. This study aims at filling gaps in the availability of consolidated data on biotoxin contamination in seafood by gathering existing data in the Mediterranean Sea. A literature review on seafood contamination by algal toxins allowed us to create a database that integrates information from various sources. This database encompasses information sourced from existing databases and 100 scientific publications. The majority of the publications pertain to lethal paralytic shellfish poisoning and diarrheic shellfish poisoning toxins, predominantly detected in mollusks, with values above the safety standards recommended by CODEX Alimentarius. This database also highlights differences concerning the main toxins involved between the regional seas of the Mediterranean basin but also in terms of data availability. This study proposes an updated assessment of human exposure to biotoxins and stresses the potential risk of exposure to biotoxins in the absence of effective food monitoring measures. It also permits us to highlight gaps and contributes to enhancing our understanding of the levels of contamination, which is crucial for promoting food safety, food security and sustainable economies in Mediterranean countries.

**Keywords:** food safety; phycotoxin; seafood contamination

### 5.4. Mambalgin-ASIC1a, a Novel Complex to Detect Non-Small-Cell Lung Cancer


**Romain Baudat ^1,^*, Évelyne Benoit ^1^, Pascal Kessler ^1^, Benoît Jego ^2^, Mathilde Keck ^1^, Vincent Thomas De Montpreville ^3^, Charles Truillet ^2^, Denis Servent ^1^**


^1^ Université Paris-Saclay, CEA, Institut des sciences du vivant Frédéric Joliot, Département Médicaments et Technologies pour la Santé (DMTS), Service d’Ingénierie Moléculaire pour la Santé (SIMoS), EMR CNRS/CEA 9004, Gif-sur-Yvette, France.^2^ Université Paris-Saclay, CEA, Institut des Sciences du Vivant Frédéric Joliot, Service Hospitalier Frédéric Joliot (SHFJ), Laboratoire D’imagerie Biomédicale Multimodale Paris Saclay (BioMaps), Orsay, France.^3^ Service D’anatomie et de Cytologie Pathologiques, Hôpital Marie-Lannelongue, Le Plessis-Robinson, France.

***** Correspondence: romain.baudat@cea.fr

**Abstract:** Non-small-cell lung cancer (NSCLC) is one of the deadliest forms of cancer, with only 16% of cases detected at an early stage. Current imaging methods, such as radiography, computed tomography and positron emission tomography, are widely used to visualize lung cancer but have certain limitations. Therefore, there is a critical need for the development of new imaging probes that are more specific to molecular targets for early diagnosis. Tumor progression is associated with various processes, notably the acidification of the microenvironment. This acidic environment disrupts ion homeostasis, including pH-sensitive channels like acid-sensing ion channels (ASICs) which are overexpressed and involved in tumor invasion, progression and metastasis. The present work reveals that ASIC1a gene expression was elevated in the A549 lung cancer cell line compared with the healthy lung cell line BEAS-2B. Furthermore, in 18 out of 24 NSCLC patients, ASIC1a was genetically overexpressed in the tumor compared to surrounding healthy tissues. To target ASIC1a, we performed the chemical synthesis of mambalgin-1 (Mamb-1), a peptide from the venom of the snake *Dendroaspis polylepis* that has a high affinity for ASIC1a. Using “click” chemistry, Mamb-1 was labeled with various fluorescent or radioactive probes. Electrophysiological studies indicated that the affinity of cyanine 5.5 (Cy-5.5)-Mamb-1 for ASIC1a remained almost unchanged compared to the unlabeled peptide. Furthermore, quantitative microscopy imaging of biopsies from 24 NSCLC patients, labeled with Mamb-1-Cy5.5, showed significant fluorescence in the tumor tissue, while no specific labeling was detected in the surrounding healthy tissue. The specificity of Mamb-1-Cy5.5 for tumor tissue was further validated by the addition of unlabeled Mamb-1 which abolished the fluorescence signal. These results, altogether, strongly suggest that fluorescent or radioactive Mamb-1 could be a valuable tool to detect ASIC1a overexpression in NSCLC, not only for early diagnosis but also for monitoring different stages of the disease.

**Keywords:** acid-sensing ion channel; cancer diagnostic; mambalgin-1; non-small-cell lung cancer

### 5.5. A New Kv1.3 Channel Blocker from the Venom of the Ant Tetramorium bicarinatum


**Guillaume Boy ^1^, Elsa Bonnafé ^1^, Laurence Jouvensal ^2^, Karine Loth ^2^, Françoise Paquet ^2^, Michel Treilhou ^1^, Arnaud Billet ^1,^***


^1^ BTSB-EA 7417, Université de Toulouse, Institut National Universitaire Jean-François Champollion (INUC), place de Verdun, 81012 Albi, France.^2^ Centre de Biophysique Moléculaire, CNRS UPR 4301, 45071 Orléans, France.

***** Correspondence: arnaud.billet@univ-jfc.fr

**Abstract:** Venoms of many species are extensively studied for human health applications such as the development of antivenoms or the identification of new pharmacological agents and therapeutic molecules. Nevertheless, ant venoms have been little studied due to the difficulty of access and the small quantities available, even though over 11,000 venomous ant species have been reported, making ant venom a major source of bioactive molecules. Our research group determined the venom composition of several ant species and now focuses on proteinaceous ant venoms. Recently, we identified the peptide Tb11a from the ant *Tetramorium bicarinatum*. This peptide features a compact helix ring structure stabilized by a disulfide bridge. Tb11a is non-cytotoxic on insect cells and exhibits paralytic activities in vivo on insects [1]. Preliminary membrane potential perturbation assays and the presence in the Tb11a of a functional dyad (Lys-Tyr), found in many voltage-gated potassium channel (Kv) blockers derived from venoms, suggest that Tb11a may interact with Kv channels. To test this hypothesis, different human potassium channels were heterologously expressed in the human embryonic kidney cell line HEK293T, and the patch-clamp technique (voltage clamp, whole-cell configuration) was used to assess the effect of Tb11a on channel activity. We found that Tb11a decreases Kv1.3 maximum current by 50%. Further experiments are ongoing to test if Tb11a modulates the activity of other Kv channels. Since some Kv channels are involved in cell cycle regulation and could be linked to carcinogenesis, we also aimed to evaluate the effects of Tb11a on some cancer processes. Breast cancer cells (MDA-MB-231) were used to test the effect of Tb11a on cell migration by wound healing and on cell proliferation by cell counting. While no effect on cell migration was detected, Tb11a had a slight but significant inhibitory effect on cell proliferation, without showing any cytotoxicity at high concentrations. We now plan to test analogs of the Tb11a peptide found in other ants to initiate a structure–function relationship study and better characterize the interaction between the peptide and the voltage-dependent potassium channel. The identification and characterization of new pharmacological agents targeting Kv channels could be of great interest for cancer research and therapy.

**Keywords:** ant venom; ion-channel blocker; potassium channel


**Reference**


Barassé, V.; Jouvensal, L.; Boy, G.; Billet, A.; Ascoët, S.; Lefranc, B.; Leprince, J.; Dejean, A.; Lacotte, V.; Rahioui, I.; Sivignon, C.; Gaget, K.; Ribeiro Lopes, M.; Calevro, F.; Da Silva, P.; Loth, K.; Paquet, F.; Treilhou, M.; Bonnafé, E.; Touchard, A. Discovery of an insect neuroactive helix ring peptide from ant venom. *Toxins (Basel)* **2023**, *15*, 600.

### 5.6. Methods for Rapid Detection of Environmental Natural Toxins

**Katrina Campbell *** 

Institute for Global Food Security, School of Biological Sciences, Queen’s University Belfast, UK.

***** Correspondence: katrina.campbell@qub.ac.uk

**Abstract:** Natural toxins and their sources in agri-food and aquaculture production are important yet complex issues. A huge investment in time and effort is placed on monitoring activities for these toxins by regulatory and industrial laboratories. Although sophisticated techniques such as chromatography and spectrometry provide accurate and conclusive results, screening tests allow a lower cost and higher throughput of samples with less operator training. Biosensors combine a biological recognition element with a transducer to produce a measurable signal proportional to the extent of interaction between the recognition element and contaminant. The different uses of biosensing instrumentation available are extremely varied, with agri-food and aquaculture analysis illustrating emerging applications. However, enzyme-linked immunosorbent assay (ELISA) and lateral flow tests dominate this market for natural toxins including marine toxins and mycotoxins as difficulties remain in combining sample preparation for feed/food contaminant analysis with biosensor technology for point-of-site testing. Nonetheless, the advantages offered by biosensors over traditional immunoassay screening methods with respect to food analysis, include automation, improved reproducibility, speed and real-time analysis. The miniaturization of immunoassays and biosensors toward nanosensing offers enhanced sensitivity, portability and multiplexing capabilities. Increased demands from stakeholders and consumers to improve food integrity illustrate a need for new tools including smart nanotechnologies for sample preparation and analysis. The aim of this presentation is to show the progress that has been made in the development and validation of nanoarrays as next-generation lateral flow arrays to be fit for purpose for the detection of natural toxins to offer an interchangeable and holistic approach to food and health safety. ELISA spot and planar waveguide technologies have been developed for the rapid and multiplex analysis of marine biotoxins, cyanotoxins, mycotoxins and plant toxins to be compatible with food control procedures. Combining advances in sample preparation tools, portable nanosensors and remote connectivity offers solutions for improved food security.

**Keywords:** cyanotoxin; detection tool; marine toxin; mycotoxin; plant toxin

### 5.7. Development of an Innovative Magnetic-Bead-Based Methodology to Enhance Antivenomics Capabilities


**Thomas Crasset ^1,^*, Damien Redureau ^1^, Fernanda Gobbi Amorim ^1^, Dominique Baiwir ^2^, Gabriel Mazzucchelli ^2^, Renaud Vincentelli ^3^, Christiane Berger-Schaffitzel ^4^, Loïc Quinton ^1^**


^1^ Mass Spectrometry Laboratory, MolSys Research Unit, University of Liège, 4000 Liège, Belgium.^2^ GIGA Proteomics Facility, University of Liège, 4000 Liège, Belgium.^3^ Laboratoire Architecture et Fonction des Macromolécules Biologiques (AFMB), Aix-Marseille Université, CNRS, 13009 Marseille, France.^4^ School of Biochemistry, University of Bristol, Biomedical Sciences Building, University Walk, Bristol BS8 1TD, UK.

***** Correspondence: thomas.crasset@uliege.be

**Abstract:** Snake envenomation remains a significant public health crisis in Africa, the Middle East, Asia and subtropical regions, mainly affecting rural populations. Each year, between 80,000 and 138,000 people die from snakebite envenomation, while three times as many survivors are left with long-term disabilities despite treatment. The main available treatments are antivenoms, which are sera-containing immunoglobulins G (IgGs) or IgG fragments targeting venom toxins. Such IgGs are purified from the blood of hyperimmunized horses or sheep. While antivenoms save many lives, they also come with significant limitations. Purified sera contain not only toxin-specific IgGs but also IgGs unrelated to venom exposure. Unfortunately, these non-specific IgGs may trigger severe adverse effects for people already in a critical situation. Moreover, antivenoms are thermally unstable and are difficult to store in regions where they are most needed. To address these challenges, the European ADDovenom project (2020–2025), funded by the European Commission (FET-Open), aims to develop a novel generation of antivenom based on ADDomers construct. ADDomers are thermally stable megadalton virus-like particles, produced at low cost and featuring 60 high-affinity binding sites, offering a promising alternative to conventional antivenoms. Here, we present a novel method for antivenomics, providing a fully automatable, faster process. Magnetic beads were coupled with EchiTab G, a monospecific antivenom targeting *Echis ocellatus* venom. The beads were incubated with *E. ocellatus* venom, and both the eluate and supernatant were analyzed using LC-MS and tryptic digestion to identify and quantify the specific toxins immuno-recognized by EchiTab G. Cross-reactivity of the antivenom was also assessed with *E. leucogaster* and *E. coloratus* venoms. This innovative method reduces the need for large amounts of solvent and valuable biological materials (venoms and antivenoms). Unlike the conventional method using NHS-activated sepharose matrices and UV-visible detection, which is impractical for small-scale analysis, our method using magnetic beads and LC-MS significantly improves sensitivity and scalability. Additionally, a novel relative quantification method for toxins in venoms, based on the three most intense, unique ions from tryptic digestion, enables quantitative measurement of the different toxin families captured by the beads. Once the methodology is optimized for traditional antivenoms, it will be used to assess the efficacy of ADDobody—firstly against synthetic phospholipases A2 targeted by ADDobodies and secondly against *E. coloratus* raw venom. Finally, the efficacy of the ADDomers structures will be evaluated. This method could accelerate antivenom development, making it more efficient, adaptable and industry ready.

**Keywords:** antivenomics; liquid chromatography; mass spectrometry; method development; snake venom

### 5.8. Clostridioides difficile Binary Toxin CDT Induces Biofilm-like Persisting Microcolonies


**Jazmin Meza Torres ^1^, Jean-Yves Tivenez Nevez ^2^, Aline Crouzols ^1^, Héloïse Mary ^3^, Minhee Kim ^3^, Lise Hunault ^4^, Susan Chamorro-Rodriguez ^1^, Emilie Lejal ^5^, Pamela Altamirano-Silva ^6^, Déborah Groussard ^7^, Samy Gobaa ^3^, Johann Peltier ^5^, Benoît Chassaing ^8,9^, Bruno Dupuy ^1,^***


^1^ Pathogenèse des Bactéries Anaérobies, Département de Microbiologie, Institut Pasteur, Université Paris-Cité, CNRS UMR6047, Paris, France.^2^ Hub d’Analyse d’Images, Département de Biologie Cellulaire et Infection, Institut Pasteur, Université Paris Cité Paris, France.^3^ Plateforme Biomatériaux et Microfluidiques, Département de Biologie du Développement et Cellules Souches, Institut Pasteur, Université Paris Cité, Paris, France.^4^ Anticorps en Thérapie et Pathologie, Département d’Immunologie, Institut Pasteur, Paris, France.^5^ Institut de Biologie Intégrative de la Cellule (I2BC), Université Paris-Saclay, CEA, CNRS, Gif-sur-Yvette, France.^6^ Centre de Recherche sur les Maladies Tropicales, Université du Costa Rica, San José, Costa Rica.^7^ Animalerie Centrale, Institut Pasteur, Paris, France.^8^ Interactions Microbiome-Hôte, Département de Microbiologie, Institut Pasteur, Université Paris Cité, Inserm U1306, Paris, France.^9^ Microbiote des Muqueuses Dans les Maladies Inflammatoires Chroniques, Inserm U1016, CNRS UMR8104, Université Paris Cité, Paris, France.

***** Correspondence: bdupuy@pasteur.fr

**Abstract:** *Clostridioides difficile*, a Gram-positive, strictly anaerobic, spore-forming bacterium, is the leading cause of antibiotic-associated diarrhea in adults. Clinical symptoms of *C. difficile* infection range from simple diarrhea to life-threatening pseudomembranous colitis. The high rate of recurrence due to relapse or reinfection is a major challenge in the management of *C. difficile* infections. Several studies have shown a close correlation between the production of the binary toxin CDT by clinical strains of *C. difficile* and higher rates of relapse and increased virulence. Although the mechanism of action of binary CDT toxin on host cells is known, its exact contribution to *C. difficile* infections has yet to be established. To understand the physiological role of binary CDT toxin during *C. difficile* infections, we established two hypoxic intestinal models, i.e., the Transwel and the Microfluidic Intestine-on-Chip systems, adapted for *C. difficile* growth. Both models were exposed to the epidemic strain UK1 and isogenic mutant strains *tcdA-tcdB-cdtAB+/-*. We show that the presence of the binary toxin CDT induces the formation of *C. difficile* microcolonies in association with mucin, which display biofilm-like properties enhancing *C. difficile* resistance to vancomycin. In vivo, biofilm-like microcolonies were also observed in both the cecum and colon of infected mice. Our results confirm the involvement of the binary toxin CDT in the long-term colonization of *C. difficile* and suggest that binary toxin CDT-dependent 3D biofilm-like structures may be involved in the persistence of *C. difficile* in the gut and play a crucial role in *C. difficile* relapse.

**Keywords:** binary toxin CDT; biofilm; *Clostridioides difficile*

### 5.9. Predatory and Defensive Venoms of Some Textilia Conidae Cone Snails


**Zahrmina Ratibou ^1^, Valentin Dutertre ^2^, Camille Gache ^2^, Tamatoa Bambridge ^2^, Serge Planes ^2^, Nicolas Inguimbert ^2^, Sébastien Dutertre ^1,^***


^1^ IBMM, University of Montpellier, CNRS, ENSCM, 34093 Montpellier, France.^2^ CRIOBE, UAR CNRS-EPHE-UPVD 3278, University of Perpignan Via Domitia, 58 avenue Paul Alduy, 66860 Perpignan, France.

***** Correspondence: sebastien.dutertre@umontpellier.fr

**Abstract:** Cone snails are specialized carnivorous marine mollusks that use neurotoxic venom to subdue prey and defend against predators. The venoms of some fish-hunting cone snails, such as the common *Pionoconus striatus*, *Gastridium geographus* and *Chelyconus purpurascens* species, have been extensively characterized. Another clade, *Textilia*, encompasses more elusive piscivorous species, such as *T. adamsonii*, *T. floccatus* or *T. julii*. As a result of their rarity, information on their venom gland composition is scarce, and the cocktail of conotoxins injected for predation or defense purposes is unknown. In order to gain a first insight into the venom–ecology relationships in this lineage, we collected the predatory and defensive venoms of *T. bullatus* and the predatory venom of *T. adamsonii* from the Marquesas Islands. LC-MS analysis of these samples revealed distinct predatory and defensive venom profiles. Further studies, including proteo-transcriptomics and bioassays, are now needed to help close the gap in our understanding of the evolved ecological roles of the injected conotoxins.

**Keywords:** cone snail; defensive venom; predatory venom; *Textilia*

### 5.10. Advancing Esophageal Adenocarcinoma Treatment Using Animal Venoms


**Chloé Matthys ^1,^*, Lou Freuville ^2^, Alain Brans ^3^, Loïc Quinton ^2^, Jean-Pierre Gillet ^1^**


^1^ Laboratory of Molecular Cancer Biology, URPhyM, NARILIS, Faculty of Medicine, University of Namur, rue de Bruxelles, 51, 5000 Namur, Belgium.^2^ Mass Spectrometry Laboratory, MolSys Research Unit, Department of Chemistry, University of Liège, Allée du Six Août, 11—Quartier Agora, 4000 Liège, Belgium.^3^ Centre for Protein Engineering, Department of Life Sciences, University of Liège, Allée du Six Août, 13—Quartier Agora, 4000 Liège, Belgium.

***** Correspondence: chloe.matthys@unamur.be

**Abstract:** Esophageal cancer is one of the most common cancers worldwide, affecting more than 450,000 people and representing a major global cancer burden. Despite many advances in diagnosis and treatment, it is also estimated to be the sixth leading cause of cancer-related mortality with an overall 5-year survival rate that remains at 15%. Esophageal cancer comprises two main histologic subtypes namely adenocarcinoma and squamous cell carcinoma. Worldwide, squamous cell carcinoma accounts for 89% of all esophageal cancers, while adenocarcinoma accounts for only 11%, but in Australia, the UK, the USA and some Western European countries, adenocarcinoma predominates, and its incidence is rising rapidly in high-income countries, in all demographic groups, particularly among white men. One reason that may explain the dismal outcome of patients with esophageal cancer is the poor response of this tumor to chemotherapy and radiation, the mainstay of pre-surgical treatment for late stages of the disease. Immunotherapy alone or in combination is being evaluated in clinical trials and shows mixed outcomes. There is a dire need to address the mechanisms of treatment resistance in this cancer and to develop new therapeutic strategies. One alternative could be the use of venom peptides, which are gaining interest as a drug discovery strategy due to their wide range of pharmacological activities. There are more than 220,000 venomous species, representing approximately 40 million toxins, of which less than 5000 have been pharmacologically characterized. In this project, we focused on adenocarcinoma and tested six fractionated venoms from different species. The resulting fractions were individually tested on three adenocarcinoma cell lines and on immortalized normal esophageal cells. Fractions showing selective cytotoxic activity against cancer cells were further analyzed by mass spectrometry to identify the sequence of the lead toxins. Efforts are now focused on identifying their mechanisms of action.

**Keywords:** cancer; cytotoxicity; drug discovery

### 5.11. Discovery of Cytolytic Proteins (Cephalolysins) in Squid Posterior Salivary Glands


**William R. Kem ***


Department of Pharmacology and Therapeutics, University of Florida College of Medicine, Gainesville, FL 32610, USA.

***** Correspondence: wrkem@ufl.edu

**Abstract:** Octopus and cuttlefish posterior salivary gland (PSG) venoms are known to contain crab paralytic proteins called cephalotoxins [1]. One was finally isolated and sequenced [2]. We searched for similar proteins in Atlantic squid (*Doryteuthis pealeii*) *PSGs.* Our discovery of crab paralytic and vertebrate (rat and flounder) hemolytic activities was initially reported in 1980 [3]. Lysis of rat erythrocytes was completely inhibited by pre-incubation of PSG homogenates with proteases (trypsin and pronase) and disulfide-bond-reducing agents; ten-minute exposure at 100 °C also inactivated hemolytic activity. The lytic activity is unlikely to be phospholipase A2 mediated, as it occurs in the absence of calcium ions and the presence of 2 mM EDTA. A single hemolytic peak (V_e_ =1.83 V_o_) was revealed during Ultrogel Ac44 column chromatography of the *Dp* PSG homogenate. This peak contained 30 and 50 kDa proteins according to SDS-PAGE analysis. Isoelectric focusing of this fraction revealed at least two hemolytic components with mildly basic (pHs 7.2 and 7.8) isoelectric points. Although PSGs of two additional species of squid (*Doryteuthis kensaki* and *Sepioteuthis lessoniana*) contained similar amounts of hemolytic activity, PSG homogenates of two octopus species (*Eledone moschata and Octopus apollyon*) and one cuttlefish (*Sepia officinalis*) lacked cytolytic activity. Thus, the PSG cytolysins may be restricted to squids.

**Keywords:** cephalolysin; cephalopod; cephalotoxin; hemolysin; squid; *Doryteuthis*; toxin


**References**


Ghiretti, F. Cephalotoxin: the crab-paralysing agent of the posterior salivary glands of cephalopods. *Nature* **1959**, 183, 1192–1193.Ueda, A.; Nagai, N.; Ishida, M.; Nagashima, Y.; Shiomi, K. Purification and molecular cloning of SE-cephalotoxin, a proteinaceous toxin from the posterior salivary gland of cuttlefish *Sepia esculenta*. *Toxicon* **2008**, *52*, 574–581.Kem, W.; Scott, J. Partial purification and characterization of a cytotoxic protein from squid (*Loligo pealei*) posterior salivary glands (abstract). *Biol. Bull.* **1980**, *159*, 475.

### 5.12. Proteomic Insights into Bothrops leucurus Venom: A Step Toward Innovative Antivenom Strategies Using Camelid Heavy-Chain Antibodies (HCAbs)


**Rainery Monteiro De Carvalho ^1,2^, Sibele Andrade Roberto ^1,2^, Jackson Vieira Leitão ^1^, Ilka Borges Biondi ^4^, Bárbara Cibelle Soares Farias Quintela ^3^, Donat Alexander De Chapeaurouge ^3^, Anna Carolina Machado Marinho ^3^, Carla Freire Celedonio Fernandes ^3^, Nidiane Dantas Reis Prado ^1^, Soraya dos Santos Pereira ^1,2,^***


^1^ Fundação Oswaldo Cruz Rondônia, Fiocruz Rondônia, Porto Velho-RO, Brazil.^2^ Programa de pós-Graduação em Biologia Experimental, Universidade Federal de Rondônia, Porto Velho, RO, Brazil.^3^ Fundação Oswaldo Cruz Ceará, FIOCRUZ CE, Eusébio-CE, Brazil.^4^ Universidade Estadual de Feira de Santana, UEFS, Feira de Santana-Bahia, Brazil.

***** Correspondence: soraya.santos@fiocruz.br

**Abstract:** Snakebite significantly impacts public health around the world. Understanding the proteomics of venomous snake venoms and searching for new antivenoms is a strategy employed by the World Health Organisation to reduce snakebite cases by 50% by 2030. *Bothrops leucurus* is a species of snake that is endemic to the Northeastern region of Brazil. The antivenoms produced in Brazil are made from fragments of monovalent (Fab) or divalent antibody fragments (F(ab’)2). However, these antivenoms do not include the venom of *B. leucurus* in the mixture of venoms used to create serum therapy. Antibodies derived from the camelid heavy-chain antibodies (HCAbs) IgG2 and IgG3 (90 kDa) and their antigen-recognition domains called nanobodies have been explored due to their physicochemical characteristics for applications as antivenoms. In this study, we performed the proteomic characterization of *B. leucurus* venom and evaluated the neutralization capacity of purified IgGs from the serum of a Lama previously immunized with *B. leucurus* venom. For venom proteome analysis, peptides were separated via reverse-phase chromatography, and precursor masses were measured with an Orbitrap mass analyzer (Q Exactive Plus). The capacity of IgGs to neutralize the cytotoxicity of the venom in the mouse myoblast cell line, C2C12, was evaluated by quantifying the levels of released lactate dehydrogenase (LDH) and cell viability using the (3-[4,5-dimethylthiazol-2-yl]-2,5 diphenyl tetrazolium bromide) (MTT) assay. The mass spectrometry experiment identified 1810 peptides and 241 proteins from *B. leucurus* venom (False Discovery Rate (FDR) <1%). Normalized Spectral Abundance Factor indicated that the top 10 abundant proteins included bradykinin-potentiating peptides, phospholipase A2 and snake venom metalloproteinases, with other families like L-amino acid oxidases and serine proteases present in smaller amounts. In the evaluation of the potential for inhibition of cytotoxicity, IgG2 reduced cytotoxicity by more than 70%, while IgG3 induced a reduction of approximately 80%. However, IgGs did not display inhibitory potential against the cytotoxicity of phospholipase A2 isolated from *B. leucurus*. In the cell viability assay using the MTT assay, cells treated with IgG2 showed cell viability of approximately 50% compared to cells incubated with phospholipase A2 and *B. leucurus* venom. The proteomic study of snake venoms endemic to the region is important for supporting the development of new treatments, such as antivenom nanobodies and diagnostic strategies. Polyclonal HCAbs displayed potential for interaction with snake toxins, showing inhibitory potential on the cytotoxic effects of *B. leucurus* venom. Additional studies are needed to verify the participation of camelid heavy-chain antibodies as an important tool applicable in the treatment of snakebite.

**Keywords:** antivenom; HCAb; proteomics; snakebite

### 5.13. Pharmacological Insights into the Vasorelaxant Activity of Montivipera venoms: Isolation of Mb-PLA2 from M. bornmuelleri Venom


**Christina Sahyoun ^1,2^, Jacinthe Frangieh ^1,2^, Lou Freuville ^3^, Christophe Verthuy ^4^, Régine Lebrun ^4^, Loïc Quinton ^3^, Ziad Fajloun ^2,5^, César Mattei ^1^, Christian Legros ^1,^***


^1^ University of Angers, MITOVASC Laboratory, Team CarME, SFR ICAT, Inserm, CNRS, Angers, France.^2^ Lebanese University, Laboratory of Applied Biotechnology (LBA3B), Azm Center for Research in Biotechnology and Its Applications, Tripoli, Lebanon.^3^ University of Liège, Laboratory of Mass spectrometry, MolSys Research Unit, Liège, Belgium.^4^ Aix-Marseille Université, CNRS, IMM-FR3479, Marseille Protéomique, Marseille, France.^5^ Lebanese University, Department of Biology, Faculty of Sciences 3, Tripoli, Lebanon.

***** Correspondence: christian.legros@univ-angers.fr

**Abstract:** Arterial hypertension is the leading cause of cardiovascular diseases and a major contributor to premature death worldwide. Despite available treatments, only 21% of patients have their blood pressure under control, highlighting the need for new therapies. In this context, our study aims to identify and characterize anti-hypertensive components from snake venoms. Indeed, hypotension is a prominent clinical manifestation upon Viperid snakebites. Hypotensive molecules identified in snake venoms induce arterial vasorelaxation and consequently reduce systemic vascular resistance and arterial pressure. Vipers of the *Montivipera* genus are endemic to the Near and Middle East regions. Their venoms are poorly studied, and their vasorelaxant effects have not been evaluated yet. Thus, we chose to screen the venoms of five *Montivipera* species, *M. bornmuelleri*, *M. bulgardaghica*, *M. albizona*, *M. xanthina* and *M. raddei*, for their vasorelaxant effects. We initiated our screening assay with the Lebanese viper, *M. bornmuelleri*, because we previously showed that its venom induces potent vasorelaxation of rat aorta rings [1]. Our results revealed that these five *Montivipera* venoms induce significant vasorelaxant effects on isolated mesenteric arteries through different endothelial pathways (including nitric oxide, cyclooxygenase and endothelium-derived hyperpolarizing factor pathways), suggesting the presence of molecules with varying modes of action. Then, using bioassay-guided fractionation, from *M. bornmuelleri* venom, we isolated a new phospholipase A2 (PLA2), named Mb-PLA2, displaying vasorelaxant effects via only endothelium-dependent pathways. Mb-PLA2 consists of 121 amino acid residues with a molecular mass of 13,6 kDa. A 3D model of Mb-PLA2 obtained by homology modeling revealed that it shares structural features with other PLA2s of the same group. The exact vasorelaxant mode of action of Mb-PLA2 and its molecular targets are yet to be identified. However, we suggest the involvement of both catalytic and non-catalytic activities of Mb-PLA2 in the observed vasorelaxant effects. In conclusion, our study evidenced the heterogeneity of vasorelaxant molecules in *Montivipera* venoms. Isolating these molecules could provide various isoforms to study vasorelaxant pathways. Additionally, the specific activity of Mb-PLA2 on the endothelium suggests a novel mode of action for this class of molecules. Therefore, identifying its molecular target(s) could provide valuable insights for developing anti-hypertensive treatments.

**Keywords:** endothelium; *Montivipera* venom; *Montivipera bornmuelleri*; phospholipase A2; vasorelaxant effect


**Reference**


Accary, C.; Hraoui-Bloquet, S.; Sadek, R.; Alameddine, A.; Fajloun, Z.; Desfontis, J.C.; Mallem, Y. The relaxant effect of the *Montivipera bornmuelleri* snake venom on vascular contractility. *J. Venom Res.* **2016**, *7*, 10–15.

### 5.14. Mechanism of Acid-Sensing Ion Channel Modulation by the Pain-Relieving Peptide Mambalgin


**Miguel Salinas ^1,^*, Pascal Kessler ^2^, Dominique Douguet ^1^, Daad Sarraf ^2^, Nicolo Tonali ^2,3^, Robert Thai ^2^, Denis Servent ^2,^*, Éric Lingueglia ^1,^***


^1^ Université Côte d’Azur, CNRS, Inserm, IPMC, LabEx ICST, FHU InovPain, France.^2^ Université Paris Saclay, CEA, Département Médicaments et Technologies pour la Santé (DMTS), SIMoS, 91191 Gif-sur-Yvette, France.^3^ CNRS, BioCIS, Université Paris-Saclay, 92290 Châtenay-Malabry, France.

***** Correspondence: salinas@ipmc.cnrs.fr (M.S.); denis.servent@cea.fr (D.S.); lingueglia@ipmc.cnrs.fr (É.L.)

**Abstract:** Acid-sensing ion channels (ASICs) are proton-gated cationic channels involved in pain and other processes, underscoring the potential therapeutic value of specific inhibitors such as the three-finger toxin from snake venom, mambalgin-1 (Mamb-1). A low-resolution structure of the human-ASIC1a/Mamb-1 complex obtained by cryo-electron microscopy was recently reported, implementing the structure of the chicken-ASIC1/Mamb-1 complex previously published. To obtain a detailed picture at the level of side-chain interactions of the binding of Mamb-1 on rat ASIC1a channels and of its inhibition mechanism, we combined structure–activity relationships of both the rat ASIC1a channel and the Mamb-1 toxin with a molecular dynamics simulation. Fingers I and II of Mamb-1, but not the core of the toxin, are required for interaction with the thumb domain of ASIC1a, and Lys-8 of finger I potentially interacts with Tyr-358 in the thumb domain. Mamb-1 does not interfere directly with the pH sensor, as previously suggested, but locks, by several contacts, a key hinge between helices α4 and α5 in the thumb domain of ASIC1a to prevent channel opening. Our results provide an improved model of inhibition of mammalian ASIC1a channels by Mamb-1 and clues for further development of optimized ASIC blockers.

**Highlights:** Fingers I and II of Mamb-1 are required for interaction with the rASIC1a thumb domain. Residue K8 in Mamb-1 finger I interacts with Y358 in the rASIC1a thumb domain. There is no apparent contact between the toxin core and the lower part of the thumb domain. Mamb-1 does not act directly on the pH sensor but on the thumb domain. Locking the hinge between helices α4 and α5 in the thumb domain prevents channel opening.

**Keywords:** acid-sensing ion channel (ASIC); inhibition mechanism; mambalgin; pain; snake toxin; sodium channel

### 5.15. L-Amino Acid Oxidase from Calloselasma rhodostoma Venom, Cr-LAAO, Induces FPR-Dependent Vital NETosis in Human Neutrophils


**Mauro Valentino Paloschi ^1^, Aleff Ferreira Francisco ^2^, Milena Daniela Souza Silva ^1^, Charles Nunes Boeno ^1^, Hallison Mota Santana ^1^, Braz Junior Campos Farias ^1^, Carolina Pereira Da Silva ^1^, Sulamita da Silva Setúbal ^1^, Ho Phin Chong ^3,4^, Choo Hock Tan ^3,5^, Juliana Pavan Zuliani ^1,6,^***


^1^ Laboratório de Imunologia Celular Aplicada à Saúde, Fundação Oswaldo Cruz, FIOCRUZ Rondônia, Porto Velho-RO, Brazil.^2^ Laboratório de Biotecnologia de Proteínas e Compostos Bioativos (LABIOPROT), Fundação Oswaldo Cruz, FIOCRUZ Rondônia, Porto Velho-RO, Brazil.^3^ Department of Pharmacology, Faculty of Medicine, University of Malaya, Kuala Lumpur, Malaysia.^4^ Faculty of Pharmacy and Biomedical Sciences, MAHSA University, Selangor, Malaysia.^5^ School of Medicine, College of Life Sciences and Medicine, National Tsing Hua University, Hsinchu, Taiwan.^6^ Departamento de Medicina, Universidade Federal de Rondônia, UNIR, Porto Velho, RO, Brazil.

***** Correspondence: juliana.zuliani@fiocruz.br

**Abstract:** Previous research has shown that L-amino acid oxidase from *Calloselasma rhodostoma* venom (Cr-LAAO) activates neutrophils, leading to processes such as chemotaxis, phagocytosis, production of reactive oxygen species (ROS) and release of pro-inflammatory cytokines and lipid mediators. Cr-LAAO also activates the nicotinamide adenine dinucleotide phosphate (NADPH) oxidase complex, contributing to ROS production that triggers the NLR family pyrin domain containing 3 (NLRP3) inflammasome, resulting in IL-1β synthesis and release. Additionally, Cr-LAAO induces dsDNA release by neutrophils, suggesting it initiates the formation of neutrophil extracellular traps (NETs) in a process known as NETosis. NETs, composed of chromatin, nuclear, granule and cytosolic proteins, play a critical role in inflammation. This study aimed to investigate the role of ROS generated by NADPH oxidase and NLRP3 inflammasome activation in Cr-LAAO-induced NETosis, which is dependent on formyl peptide receptor 1 (FPR1). Using a microarray gene expression assay, we identified genes involved in the NET formation pathway. The fold change matrix analysis revealed both upregulation and downregulation of gene expression in Cr-LAAO-stimulated neutrophils compared to non-stimulated neutrophils, with 104 genes selected for their relevance in NET formation. Immunofluorescence assays confirmed NET formation in Cr-LAAO-stimulated neutrophils. Quantification of dsDNA released by neutrophils showed a significant increase induced by Cr-LAAO, which was reduced in the presence of FPR1, ROS and NLRP3 inhibitors. The viability of neutrophils was not affected by either Cr-LAAO or the inhibitors, indicating that Cr-LAAO stimulates a vital form of NETosis. This study is the first to demonstrate the influence of FPR1, ROS produced by both NADPH oxidase and NLRP3 inflammasome activation on the FPR1-dependent NETosis process induced by Cr-LAAO.

**Support:** FAPERO, PROEP-FIOCRUZ-RO, CNPq, CAPES

**Keywords:** CR-LAAO; L-amino acid oxidase; neutrophil; neutrophil extracellular trap (NET); snake venom

## 6. Poster Presentations (When More than One Author, the Underlined Name Is That of the Presenter)

### 6.1. Development of an Electrochemical Model for the Detection and Evaluation of the Potential Toxicity of Naturally Occurring Furanic Compounds


**Imène Ayaden ^1,^*, Céline Hoffmann ^2^, Chouaha Bouzidi ^1^, Thomas Gaslonde ^1^, Joëlle Perard ^1^, Florence Souquet ^1^, Xavier Cachet ^1^**


^1^ Cibles Thérapeutiques et Conception de Médicaments CiTCoM, UMR8038, CNRS, Faculté de Pharmacie de l’Université Paris Cité, Paris, France.^2^ Unité de Technologies Chimiques et Biologiques pour la Santé (UTCBS) Inserm U1267 CNRS UMR8258, Faculté de Pharmacie de l’Université Paris Cité, Paris, France.

***** Correspondence: imeneayaden@gmail.com

Abstract: Please refer to Section 5.1.

### 6.2. A New Peptide from the Spider Poecilotheria subfusca Is a Potent Blocker of the Human Cav1.2 Channel


**Tânia C. Gonçalves ^1,2^, Michel De Waard ^3,4,5^, Michael Kurz ^6^, Stephan De Waard ^3,4,5^, Camille Sanson ^1^, Jérôme Montnach ^3,4,5^, Françoise Chesney ^1^, Rémy Béroud ^3^, Denis Servent ^2^, Michel Partiseti ^1^, Jean-Marie Chambard ^1^, Évelyne Benoit ^2,^***


^1^ Sanofi R&D, Integrated Drug Discovery—In Vitro Biology, F-94440 Vitry-sur-Seine, France.^2^ Université Paris-Saclay, CEA, Institut des Sciences du Vivant Frédéric Joliot, Département Médicaments et Technologies pour la Santé (DMTS), Service d’Ingénierie Moléculaire pour la Santé (SIMoS), EMR CNRS/CEA 9004, F-91191 Gif-sur-Yvette, France.^3^ Smartox Biotechnology, 6 rue des Platanes, F-38120 Saint-Egrève, France.^4^ Nantes Université, CNRS, Inserm, l’institut du thorax, F-44000 Nantes, France.^5^ LabEx “Ion Channels, Science & Therapeutics”, F-06560 Valbonne, France.^6^ Sanofi R&D, Integrated Drug Discovery—Synthetic Molecular Design, G-65929 Frankfurt, Germany.

***** Correspondence: evelyne.benoit@cea.fr

**Abstract:** Over the last two decades, animal venom toxins have been explored as an original source of new antinociceptive drugs targeting ion channel subtypes. Although the gold standard remains the time-consuming manual patch-clamp, automated patch-clamp platforms were developed to increase the number of positive hits. In the present study, we performed high-throughput screening of a collection of venom toxins through automated whole-cell patch-clamp experiments on human embryonic kidney (HEK)-293 cells overexpressing the genetically validated antinociceptive target Nav1.7 or the cardiac Nav1.5 human subtypes of voltage-gated sodium (hNav) channels. This first step aimed to identify bioactive peptides that were then purified, sequenced and chemically synthesized. The second step consisted in characterizing the synthetic peptide of interest on hNav, voltage-gated potassium (hKv) and calcium (hCav) and inward-rectifier potassium (hKir) channel subtypes overexpressed in cells, and on the action potential and calcium currents of human induced pluripotent stem cell-derived cardiomyocytes. The key results are the identification of poecitoxin-1a, the first toxin to be isolated from the *Poecilotheria subfusca* spider venom, and its characterization as a peptide of 35 amino acid residues belonging to the inhibitor cystine knot (ICK) structural family. This peptide was found to inhibit hCav1.2 with high affinity (a 24 nM dose inducing 50% current inhibition) and high selectivity compared to other hCav subtypes. In conclusion, poecitoxin-1a is the first ICK spider toxin specifically targeting hCav1.2. This peptide represents, at least, a valuable tool to study the Cav1.2 function and location and, at best, a valuable drug to treat some cardiomyopathies such as the long QT syndrome.

**Keywords:** Cav1.2 channel subtype; human voltage-gated ion channel subtype; poecitoxin-1a; spider peptide

### 6.3. High-Throughput Screening of Venoms on Nav1.4: A Therapeutic Benefit


**Floriane Bibault ^1,4,^*, Barbara Ribeiro ^1,4^, Jérôme Montnach ^1,4^, Hugo Millet ^1,4^, Lucie Jaquillard ^2^, Rémy Béroud ^2^, Sophie Nicole ^3,4^, Michel De Waard ^1,4^**


^1^ L’institut du Thorax, Inserm UMR1087/CNRS UMR6291, Nantes, France.^2^ Smartox Biotechnology, Saint-Egrève, France.^3^ Institut de Génomique Fonctionnelle, Inserm UMR U1191, CNRS UMR 5203, UM, Montpellier, France.^4^ LABEX Ion Channel Science and Therapeutics (ICST).

***** Correspondence: floriane.bibault@univ-nantes.fr

**Abstract:** Human sodium channels Nav1.4 are heterodimers with a pore-forming α-subunit encoded by the *SCN4A* gene and expressed only in skeletal muscle fibers, where they play a key role in muscle contraction. Mutations in *SCN4A*, whether loss-of-function or gain-of-function, lead to an imbalance in ion exchange and cause congenital muscle weakness or myotonia. To date, there is no treatment available for muscle weakness—some Nav blockers are used in the treatment of myotonia, such as mexiletine, but there is no selective activator or inhibitor of Nav1.4. This is why it is essential to find new molecules that specifically target this channel. Since ion channels are considered the main targets of natural venoms, an approach based on venom library screening was used to identify generic and specific modulators of human Nav1.4. Eighteen spider venoms from different parts of the world were fractionated into 64 fractions each by reversed-phase high-performance/pressure liquid chromatography (HPLC). For each venom, we selected the 46 most hydrophobic fractions and distributed them into a 384-well microplate, i.e., 8 wells per fraction and 16 control wells. Each of the 18 plates was then screened using a high-throughput patch-clamp on human embryonic kidney cell lines stably expressing the human Nav1.4 channel. Automated analysis of the activation and inactivation protocols, using R language, enabled us to identify 19 activators and 21 inhibitors from the 828 fractions screened. We observed that the activating fractions modified the current amplitude or slowed the kinetics of inactivation, while the inhibiting fractions modified only the current amplitude. These are known activation or inhibition mechanisms for spider venoms acting on other sodium channels. These interesting fractions were re-fractionated by cation-exchange HPLC and then screened again by high-throughput patch-clamp. Finally, the final, active fractions were sent for sequencing by mass spectrometry to identify the peptides of interest. This work represents the first large-scale screening of active toxins toward human Nav1.4. The selected peptides will also be tested for their selectivity toward Nav1.4 compared to other Nav subtypes. With this work, we hope to describe new Nav1.4-selective toxins with potential development for the treatment of sodium muscle channelopathies.

**Keywords:** high-throughput; Nav1.4; patch-clamp; venom library screening

### 6.4. The Venom of Macrovipera lebetina lebetina: Exploring Its Cytotoxic and Antibacterial Effects


**Noémie Brial ^1^, Tolis Panayi ^2,3^, Vicky Nicolaidou ^2,^*, Koulla Achilleos ^2^, Yiannis Sarigiannis ^3,^***


^1^ Department of Biology, University of Angers, Angers, France.^2^ Department of Life Sciences, University of Nicosia, Nicosia, Cyprus.^3^ Department of Health Sciences, University of Nicosia, Nicosia, Cyprus.

***** Correspondence: nicolaidou.v@unic.ac.cy (V.N.); sarigiannis.i@unic.ac.cy (Y.S.)

**Abstract:** Venoms are rich sources of compounds ranging from small organic molecules and peptides to proteins, which have attracted researchers’ interest to find new chemical compounds with therapeutic applications. The snake *Macrovipera lebetina lebetina* is an endemic Cypriot subspecies whose venom has not been widely studied, so its precise composition remains to be determined. Here, we present the current bioassays performed with the crude venom on the human cell lines, HEK293T and MDA-MB231, which already highlight promising results regarding the cytotoxicity of the venom and its anticancer potential. Furthermore, the crude venom was tested against *Escherichia coli*, *Staphylococcus epidermidis*, *Streptococcus pneumoniae* and *Neisseria subflava* with significant results, which demand further investigation. These results could not only deepen our understanding of the composition of *M. lebetina lebetina* venom, but also pave new pathways for the development of innovative therapeutic treatments. In addition to our studies with the crude venom, we further identified several peptides. Among them was a characteristic pGlu-Lys-Trp tripeptide found in most *Macrovipera* venoms. This peptide was synthesized in both its native and amide forms, and it is currently being studied for its physicochemical properties, by exploring its role in the venom.

**Keywords:** antibacterial; anticancer; bioassay; snake venom

### 6.5. On the Intriguing World of Snake Venom Toxins: Unraveling the Catalytic Pathway of Phospholipase A2 Toxin in a Membrane Model


**Juliana Castro-Amorim *, Alexandre Pinto, Maria João Ramos, Pedro A. Fernandes**


LAQV-REQUIMTE, Department of Chemistry and Biochemistry, Faculty of Sciences, University of Porto, Porto, Portugal.

***** Correspondence: up201505227@fc.up.pt

**Abstract:** Snake-venom-secreted phospholipases A2 (svPLA2s) are potent toxic enzymes found in nearly all snake venoms. Upon envenomation, these enzymes compromise the integrity of plasma membranes by hydrolyzing phospholipids, resulting in severe pathological effects such as myonecrosis, hemotoxicity, blistering, inflammation and pain. Despite the critical role of svPLA2s in the toxic process of snakebites, the exact reaction mechanism remains inadequately understood, limiting the advancement of effective antivenoms. This study centers on myotoxin-I, a svPLA2 enzyme present in the venom of the Central American pit viper, Terciopelo (*Bothrops asper*)—a species notorious for its wide distribution, aggressivity, and responsibility for severe envenomation events in its habitat. We explored the intricate interaction between svPLA2 and a 1:1 phosphatidylcholine/phosphatidylserine membrane model, examining how enzyme binding alters membrane structure and dynamics. Moreover, we investigated two reaction mechanisms for svPLA2s: the “single-water mechanism” and the “assisted-water mechanism”, previously proposed for the homologous, non-toxic human PLA2. Through umbrella sampling simulations at the PBE/MM level of theory, our findings suggest that while both mechanisms are plausible, the “assisted-water mechanism” is more favorable, featuring a lower activation-free energy barrier (21.84 kcal/mol) during the hydrolysis of the phosphatidylcholine substrate. The conserved architecture of the svPLA2 active site across species implies a shared catalytic mechanism, likely extending to most viperid species. Furthermore, our study demonstrates that the only small-molecule inhibitor currently in clinical trials for snakebites serves as an exemplary transition state analog. This insight underscores the importance of elucidating the catalytic mechanisms of snake venom sPLA2s in the quest to develop novel, effective inhibitors.

**Keywords:** enzymatic mechanism; phospholipase A2; snake venom; toxin; umbrella sampling

### 6.6. Pharmacological Screen for New Therapeutically Relevant Calcium Channel Modulators


**Leos Cmarko ^1,2,^*, Norbert Weiss ^3^, Michel De Waard ^1^**


^1^ Nantes Université, CNRS, Inserm, L’institut du Thorax, F-44000 Nantes, France.^2^ Charles University, Prague, Czech Republic.^3^ Department of Pathophysiology, Third Faculty of Medicine, Charles University, Prague, Czech Republic.

***** Correspondence: leos.cmarko@univ-nantes.fr

**Abstract:** Pharmacological modulation of voltage-gated T-type calcium channels (comprising three distinct isoforms, Cav3.1, Cav3.2 and Cav3.3) has become a promising avenue for the treatment of several neurological disorders such as chronic pain and epilepsy. To identify novel modulators of T-type calcium channels, I used automated patch-clamp electrophysiology as a high-throughput platform to screen a library of spider venom peptides. The screening of 1800 venom fractions gave rise to three previously undescribed peptides with inhibitory activity on T-type currents. Using a combination of top-down and bottom-up mass spectrometry, I was then able to resolve the sequences of these peptides. Based on one of the sequences, a synthetic peptide was produced and tested on both recombinant T-type channels and T-type channels expressed natively in dorsal root ganglion neurons. Altogether, I show that high-throughput electrophysiological screening is an effective approach for investigating new modulators of T-type calcium channels. As a result, I describe a novel peptidic toxin that acts as an inhibitor of T-type channels with an IC_50_ < 100 nM.

**Keywords:** peptide; screen; T-type channel; venom

### 6.7. Development of a MALDI-Mass Spectrometry Imaging Method to Understand the Mechanisms of Snake Envenomation


**Axel De Monts De Savasse ^1^, Virginie Bertrand ^1^, Stefanie Menzies ^2^, Nicholas Casewell ^3^, Loïc Quinton ^1,^***


^1^ Laboratory of Mass Spectrometry (MSLab), Université de Liège, Liège, Belgium.^2^ Department of Biomedical and Life Sciences, Lancaster University, Lancaster, UK.^3^ Centre for Snakebite Research and Interventions, Department of Tropical Disease Biology, Liverpool School of Tropical Medicine, Liverpool, UK.

***** Correspondence: loic.quinton@uliege.be

**Abstract:** With an estimated 81,000 to 138,000 deaths per year, snakebite envenomation has been considered a neglected tropical disease by the World Health Organization since 2017. Snake venoms are complex mixtures of peptides, proteins and enzymes that primarily target the neuromuscular and hemostatic systems, with some causing local tissue necrosis. Key toxins responsible for cytotoxicity are phospholipases A2 (PLA2s) and snake venom metalloproteinases. PLA2s (10–20 kDa) hydrolyze phospholipids, inducing cell death and necrosis at the bite site, while snake venom metalloproteinases (20–100 kDa) cleave basement membrane proteins and alter cell adhesion. Matrix-assisted laser desorption/ionization (MALDI)-mass spectrometry imaging (MSI) is a powerful technique to study the spatial distribution of many types of molecules (peptides, lipids, metabolites and even small proteins) across a slice of tissue, a whole specimen or a bacterial culture. Our study exploits MSI to pave the way for a new approach to studying snakebite envenomation and the resulting tissue alterations made by venom toxins. MALDI-MSI was used to visualize the action of PLA2s from the crude venom of *Echis leucogaster* on mouse muscle tissue as a validated proof of concept. The degradation of phospholipids into lysophospholipids likely caused by PLA2s from the venom was successfully imaged and analyzed in both positive and negative ionization modes. The perspectives are now to expand the study of other toxin activities on various tissues by testing *(i)* other snake venoms, *(ii)* purified toxins and *(iii)* venoms fractionated by high-performance/pressure liquid chromatography. Finally, an in vivo model in which snake venom is directly injected into a mouse will be developed to analyze the distribution of the venom through the skin and tissues.

**Keywords:** MALDI-MSI; PLA2; snake venom metalloproteinase; snake venom toxin

### 6.8. Affinity Capture and Direct Identification of GPCR-Binding Toxin Ligands Using Mass Spectrometry


**Lou Freuville ^1,^*, Rudy Fourmy ^2^, Aude Violette ^2^, Alain Brans ^3^, Loïc Quinton ^1^**


^1^ Mass Spectrometry Laboratory, MolSys Research Unit, Department of Chemistry, University of Liège, Allée du Six Août, 11—Quartier Agora, 4000 Liège, Belgium.^2^ Alphabiotoxine Laboratory sprl, 7911 Montroeul-au-bois, Belgium.^3^ Centre for Protein Engineering, University of Liège, 4000 Liège, Belgium.

***** Correspondence: lfreuville@uliege.be

**Abstract:** Animal venoms are rich, complex mixtures containing many biologically active peptides that exhibit high selectivity and potency against membrane receptors, such as G-protein coupled receptors (GPCRs). Venom-derived toxins provide valuable tools for exploring receptor function and have significant potential in guiding pharmacophore modeling for drug discovery. Despite the growing interest in venoms as a source of therapeutic leads, the field remains underdeveloped, largely due to the complexity of venom components and the low-throughput nature of existing screening techniques, which target multiple molecular receptors of interest. This study presents an innovative methodology combining affinity capture on cellular membranes with mass spectrometry to identify peptide toxins from venoms. Our approach specifically targets the human type 2 vasopressin receptor (hV2R), a GPCR known for binding its endogenous ligand vasopressin and the venom peptide mambaquaretin-1, isolated from *Dendroaspis angusticeps* (green mamba) venom. Cell membranes overexpressing hV2R were incubated with fractionated venom samples to achieve this. Toxins displaying some affinity for the receptor bound to the membranes, while those without affinity remained in solution. The bound toxins were then analyzed by MALDI mass spectrometry, enabling their identification directly from the mixture. Control experiments included using known hV2R ligands as positive controls and membranes not overexpressing hV2R as negative controls. Our results validate the effectiveness of this affinity-based “fishing” strategy for directly identifying receptor-binding toxins from complex mixtures like venoms. This methodology paves the way for high-throughput screening of venoms, enabling the exploration of numerous peptide candidates against a wide variety of molecular targets.

**Keywords:** affinity capture method; GPCR; mass spectrometry

### 6.9. Recombinant Production of Snake Venom Metalloproteinases: Unlocking the Potential to Develop Novel Snakebite Treatments


**Sophie Hall ^1,^*, Srikanth Lingappa ^1^, Konrad Hus ^1^, Richard Stenner ^1^, Johara Boldrini-França ^1^, Dakang Shen ^1^, Iara Aimê Cardoso ^2^, Nicholas R. Casewell ^2^, Mark Wilkinson ^2^, Maria Molina ^3^, Andrew Mumford ^3^, Imre Berger ^1^, Christiane Berger-Schaffitzel ^1^**


^1^ School of Biochemistry, University of Bristol, Bristol, UK.^2^ Centre for Snakebite Research & Interventions, Liverpool School of Tropical Medicine, Liverpool, UK.^3^ Bristol Heart Institute, University of Bristol, Bristol, UK.

***** Correspondence: sophie.hall@bristol.ac.uk

**Abstract:** Snakebite envenomation, classified by the World Health Organization as a Category A Neglected Tropical Disease, causes more than 100,000 deaths and 400,000 severe disabilities annually. Despite this, antivenom treatments have remained largely unchanged for more than a century and are moderately effective, expensive and prone to severe adverse effects. Snake venom is a complex mixture of toxins, synergistically working to induce necrosis, hemorrhage, paralysis and ultimately death in victims. In vipers, the most potent toxins are snake venom metalloproteinases (SVMPs). However, SVMP research has been severely impeded by the inability to produce these proteins recombinantly. To overcome this bottleneck, we established a baculovirus/insect cell expression protocol using the MultiBac system to successfully produce SVMPs of all three classes (PI, PII, PIII). This protocol is generally applicable to all SVMPs. Zn^2+^ addition resulted in active SVMPs which we subsequently characterized through analysis of fibrinolytic and caseinolytic activity as well as blood clotting and platelet aggregation.

**Keywords:** recombinant; SVMP; toxin

### 6.10. Targeting Phospholipases A2 in Echis Venoms Using a Novel ADDobody/ADDomer Platform


**Konrad Hus ^1^, Huan Sun ^1^, Richard Stenner ^1^, Georgia Balchin ^1^, Sophie Hall ^1^, Johara Stringari ^1^, Srikanth Lingappa ^1^, Dakang Shen ^1^, Priscila El-Kazzi ^2^, Stefanie Menzies ^3^, Iara Cardoso ^3^, Fernanda Amorim ^4^, Loïc Quinton ^4^, Rute Castro ^5^, António Roldão ^5^, Nicholas R. Casewell ^3^, Renaud Vincentelli ^2^, Imre Berger ^1^, Christiane Berger-Schaffitzel ^1,^***


^1^ University of Bristol, School of Biochemistry, Bristol, UK.^2^ CNRS/Aix-Marseille Université, Laboratoire Architecture et Fonction des Macromolécules Biologiques (AFMB), Marseille, France.^3^ Liverpool School of Tropical Medicine, Centre for Drugs & Diagnostics, Liverpool, UK.^4^ University of Liège, MolSys Research Unit, Mass Spectrometry Laboratory, Liège, Belgium.^5^ iBET, Instituto de Biologia Experimental e Tecnológica, Oeiras, Portugal.

***** Correspondence: christiane.berger-schaffitzel@bristol.ac.uk

**Abstract:** Snake venom phospholipases A_2_ (PLA_2_s) are enzymes that play a significant role in the pathophysiology of snakebite envenoming, contributing to myotoxic, hemotoxic and neurotoxic effects. Current antivenom therapies are often inadequate, with only 10–15% of antibodies effectively binding to venom toxins, underscoring the need for more effective solutions. To generate new antivenoms, we developed the ADDovenom platform, which utilizes ADDobodies as binding units. ADDobodies can be assembled into large ADDomer nanoparticles with 60 binding sites, significantly enhancing their ability to neutralize venom components. We recombinantly produced the most prevalent enzymatic PLA_2_s from *Echis* venoms using the MultiBac insect cell expression system. These PLA_2_s were used as antigens for ribosome display in vitro selections employing an ADDobody library with a diversity of ~10^12^. Selected ADDobodies were characterized by inhibition ELISA, Cayman assays (soluble substrate) and Enzchek assays (substrates in membranes). We converted ADDobody C5 into ADDomer C5, which features 60 anti-PLA_2_ binding sites. ADDomer C5 particle exhibited low nanomolar IC_50_ values in vitro and neutralized native PLA_2_s in *Echis* venoms, providing a convincing proof of concept for this platform.

**Keywords:** ADDobody; ADDomer; ADDovenom; phospholipase A2; snakebite antivenom development

### 6.11. Sodium Channel Toxins of a North Sea Anemone


**William R. Kem ^1,^*, Cecilia Sanchez ^2^, Kari Basso ^2^, Steve Peigneur ^3^, Douglas Kraft ^4^, Laszlo Beress ^5^**


^1^ Pharmacology and Therapeutics, University of Florida College of Medicine, Gainesville FL, USA.^2^ Department of Chemistry, University of Florida, Gainesville, FL, USA.^3^ Pharmaceutical Sciences, KU University of Leuven, Belgium.^4^ Sterling Research Group, Rensselaer, NY, USA.^5^ Internal Medicine, Pharis Biotech GmbH/Medical School, Hannover, Germany.

***** Correspondence: wrkem@ufl.edu

**Abstract:** Sea anemones possess a variety of peptide neurotoxins affecting ion channels and actinoporins generating membrane pores. Perhaps the most ubiquitous toxins are the ~5 kDa, 3 disulfide bond peptide neurotoxins that delay the inactivation of voltage-gated sodium channels (Navs) in nerve and muscle cells. Over the past few decades, more than 70 isotoxins have been reported that vary in sequence and the Nav subtypes they preferentially target. Comparisons of sequence with activity can guide the synthesis and testing of non-natural analogs required to reveal the basis for their Nav subtype selectivity [1]. However, more sequence data are needed to determine how these toxins have evolved over the past 500+ million years. The previously reported toxin sequences were obtained from only ~3% of the 425 known genera of sea anemones. Since sea anemone systematics is now based upon more rigorous molecular and morphological data [2], it should be possible to decipher at least some aspects of the molecular evolution of this group of neurotoxins. Here, we report our progress in chemically and pharmacologically characterizing two toxins from a northern European sea anemone [3] that have relatively unique amino acid sequences compared with previously reported sequences. The actions of these toxins were initially investigated on vertebrate nerve and muscle preparations [4]. Our recent electrophysiological analyses of the actions of these toxins indicate that at low concentrations they preferentially inhibit the inactivation of human Navs 1.2 (neuronal), 1.4 (skeletal muscle) and 1.5 (cardiac muscle) and of a cockroach neuronal Nav.

**Keywords:** sea anemone; sodium channel; toxin


**References**


Moran, Y.; Cohen, L.; Kahn, R.; Karbat, I.; Gordon, D.; Gurevitz M. Expression and mutagenesis of the sea anemone toxin Av2 reveals key amino acid residues important for activity on voltage-gated sodium channels. *Biochemistry* **2006**, *45*, 8864–8873.Rodriguez, E.; Barbeltos, M.; Brugler, M.; Crowley, L.; Grajales, A.; Gusmao, L.; Haussermann, V.; Reft, A.; Daly, M. Hidden among sea anemones: The first comprehensive phylogenetic reconstruction of the Order Actinaria reveals a novel group of hexacorals. *PLOS One* **2014**, *9*, 1–16.Beress, L.; Zwick, J. Purification of two crab-paralyzing polypeptides from the sea anemone *Bolocera tuediae*. *Marine Chem.* **1980**, *8*, 333–338.Tesseraux, I.; Gulden, M.; Schumann, G. Effects of a toxin isolated from the sea anemone *Bolocera tuediae* on electrical properties of isolated rat skeletal muscle and cultured myotubes. *Toxicon* **1989**, *27*, 201–210.

### 6.12. Structural and Functional Properties of MTX1, a New Maurotoxin Homolog Isolated from the Venom of the Scorpion Scorpio maurus palmatus


**Rym El Fessi ^1^, Oussema Khamessi ^1^, Michel De Waard ^2^, Riadh Marrouchi ^1^, Riadh Kharrat ^1,^***


^1^ Laboratory of Venoms and Laboratory of Venoms and Therapeutic Biomolecules, Pasteur Institute of Tunis, University of Tunis El Manar, 13 place Pasteur, BP74, Tunis 1002, Tunisia.^2^ L’institut du Thorax, Nantes Université, CNRS, Inserm, F-44000 Nantes, France.

***** Correspondence: riadh.kharrat@pasteur.tn

**Abstract:** Maurotoxin (MTX), purified from the scorpionid *Scorpio maurus*, is a potent ligand for potassium channels. It shows a broad specificity as being active on both voltage-gated potassium channels (Kv) and small-conductance calcium-activated potassium channels (SKCa). By following the ability to recognize specific antibodies generated against MTX, a new maurotoxin-like peptide, MTX1, was isolated from *Scorpio maurus palmatus* venom. MTX1 contains 34 amino acid residues, and its sequence shares 63% identity with that of MTX. MTX1 induced pronounced in vivo toxicity in mice, which exceeded that of MTX by a factor of eight. In vitro experiments demonstrated that MTX1 binds to rat brain synaptosomes competitively with 125I-apamin (IC50 = 1.7 nM) or 125I-charybdotoxin (IC50 = 5 nM). The MTX1 effects on three cloned Kv channels (Kv1.1, Kv1.2 and Kv1.3) were investigated in *Xenopus* oocytes using electrophysiology experiments. Like MTX, MTX1 preferably blocks Kv1.2 versus Kv1.1 and Kv1.3. Very few reports investigate the structural features that underlie the specific recognition of SKCa channels by scorpion toxins. Our computational analysis demonstrates that the beta-sheet region of MTX exerts a key role in SKCa channel recognition, rather than its alpha-helix region. On the other hand, we showed that the residue Arg27, located in the C-terminal region, plays a major role in enhancing the binding affinity of MTX1 toward Kv1.2 and SKCa channels.

**Keywords:** calcium-activated potassium channel (SKCa); potassium channel; scorpion venom; short scorpion toxin; voltage-dependent potassium channel (Kv)

### 6.13. Investigation of the Involvement of Tetramorium bicarinatum Venom Peptides in Its Innate Immunity


**Stéphanie Long *, Jean-Michel Malgouyres, Steven Ascoët, Arnaud Billet, Elsa Bonnafé**


Equipe BTSB-EA 7417, Université de Toulouse, Institut National Universitaire Jean-François Champollion (INUC), Place de Verdun, Albi, France.

***** Correspondence: layyiestephanie@gmail.com

**Abstract:** Venom peptides exhibit functional similarities with Host Defense Peptides (HDPs). Indeed, myrmicitoxins from the *Tetramorium bicarinatum* ant share cytotoxic, antimicrobial and immunomodulatory properties with already described HDPs, in relation to their role as defense molecules in the venom. However, their contribution to another defense process, the immune function, remains unexplored. Insect HDP reservoirs stand on hemocytes and fat body cells, the main actors in innate immunity, and immune responses are mediated by specific signaling pathways. Besides their synthesis in the venom gland, may venom peptides be also produced by these cells and recruited as immune effectors? To address this question, we performed RT-PCR to determine the expression patterns of the basal venom peptide genes (*vpg*) and immunity related-genes (*Ig*) in fresh fat bodies and then in fat body explants after 6 h bacterial challenge with *Bacillus subtilis* peptidoglycan. The effects of the peptides were also assessed on the *Drosophila melanogaster* hemocyte-derived cell line, S2, to evaluate their ability to induce immune pathways. We selected four biologically characterized peptides (MYRTX_A1_-Tb0a, MYRTX_A1_-Tb1a, MYRTX_A1_-Tb2a and MYRTX_A1_-Tb8a) at various times and concentrations. Their effects on immune pathways were compared to those of 2 µg/mL peptidoglycan for 6 h and 1 µM 20-hydroxyecdysone for 24 h. Low to very high basal *vpg* and *Ig* expression levels were found in fresh fat bodies, and for some of them, basal levels were increased after 6 h bacterial challenge. Nevertheless, the results still need to be confirmed because of a lack of reproducibility. Peptides at different concentrations induced expression of HDP encoding genes *Stat92E*, *upd1*, *upd3* and *TotB, relish* involved in the immune process in S2 cells after 6h exposure. These results suggest that venom peptides may activate several immune pathways, leading to phagocytosis, expression of HDPs or immune cell differentiation. The limitation of our study lies in the explant culture. Our results seem to highlight that innate immunity is closely related to venomous function in the ant. RNA-seq analysis will be carried out to deepen our findings.

**Keywords:** ant; immunity; Host Defense Peptide; *Tetramorium bicarinatum*; venom

### 6.14. Multimodal Study of the Toxicity of C17-Sphinganine Analog Mycotoxin: Impact on Vital Organs, Cytotoxicity and Genotoxicity


**Zeineb Marzougui ^1,2^, Sylvie Huet ^2^, Anne-Louise Blier ^2^, Ludovic Le Hegarat ^2^, Kevin Hogeveen ^2^, Afef Bahlous ^3^, Haïfa Tounsi ^4^, Riadh Kharrat ^1^, Valérie Fessard ^2^, Riadh Marrouchi ^1,^***


^1^ Laboratory of Venoms and Therapeutic Biomolecules, Pasteur Institute of Tunis, University of Tunis El Manar, 13 place Pasteur, B.P. 74, 1002 Tunis-Belvédère, Tunisia.^2^ Unit of Contaminant Toxicology, French Agency for Food, Environmental and Occupational Health & Safety (ANSES), 10 B rue Claude Bourgelat, 35306 Fougères, France.^3^ Laboratory of Human and Experimental Pathology, Pasteur Institute of Tunis, University of Tunis El Manar, 13 place Pasteur, B.P. 74, 1002 Tunis-Belvédère, Tunisia.^4^ Laboratory of Clinical Biochemistry and Hormonology, Pasteur Institute of Tunis, University of Tunis El Manar, 13 place Pasteur, B.P. 74, 1002 Tunis-Belvédère, Tunisia.

***** Correspondence: riadh.marrouchi@pasteur.tn

**Abstract:** The neurotoxin “C17-sphinganine analog mycotoxin (C17-SAMT)” has been identified as the cause of unusual toxicity in mussels from the Bizerte Lagoon in northern Tunisia. This toxin induces severe toxic effects in mice, including flaccid paralysis, respiratory distress and rapid death. To assess the potential health risks, a series of studies were conducted to evaluate its subchronic toxicity, genotoxicity and cytotoxicity. In subchronic toxicity trials following OECD guideline 407, mice exposed to C17-SAMT displayed significant damage to vital organs, including reduced stomach weight, inflamed intestines and signs of nephritis. Biochemical and hematological analyses revealed elevated transaminase levels, increased lactate dehydrogenase, reduced red blood cell counts and elevated leukocyte counts. Microscopic examination revealed severe damage to the heart, lungs, kidneys, stomach, colon, small intestine and liver, indicating that C17-SAMT can cause systemic organ damage, including myocardial atrophy, lung necrosis, kidney inflammation and acute hepatitis. In vitro studies using human cell lines further confirmed the toxic potential of C17-SAMT. The toxin induced an increase in micronucleus formation in TK6 cells, suggesting mutagenic effects. Cytotoxicity assays in HepaRG cells showed that C17-SAMT caused mitochondrial dysfunction, decreased ATP levels and altered the expression of key proteins involved in cellular stress responses, including superoxide dismutase (SOD2), heme oxygenase (HO-1) and NF-κB. Additionally, in vivo genotoxicity studies using comet and micronucleus assays demonstrated that C17-SAMT induced DNA damage in the liver, although no significant effects were observed in DNA oxidation or chromosomal damage in bone marrow cells. Overall, these findings highlight the significant toxic potential of C17-SAMT, particularly its ability to cause severe organ damage, mitochondrial dysfunction and DNA damage. Further research is necessary to fully understand its mode of action and the broader implications for human health.

**Keywords:** C17-sphinganine analog mycotoxin; cytotoxicity; genotoxicity; subchronic toxicity

### 6.15. Advancements in VLP-Based Antivenoms and Enhanced Venom Profiling Through the ADDovenom Project


**Damien Redureau ^1,^*, Fernanda Gobbi Amorim ^1^, Thomas Crasset ^1^, Stefanie Menzies ^2^, Nicholas R. Casewell ^2^, Loïc Quinton ^1^**


^1^ Laboratory of Mass Spectrometry, MolSys Research Unit, University of Liège, Liège, Belgium.^2^ Centre for Snakebite Research and Interventions, Liverpool School of Tropical Medicine, Pembroke Place, Liverpool, UK.

***** Correspondence: dredureau@uliege.be

**Abstract:** The development of advanced snakebite therapeutics is critical given the high morbidity and mortality rates associated with envenomation. The use of conventional animal-derived antivenoms is limited by issues of efficacy, safety and economic viability. The ADDovenom project addresses these limitations by pioneering a novel snakebite treatment based on thermostable-protein-based ADDomer nanoparticles. Derived from the adenovirus penton base protein, ADDomers present 60 high-avidity binding sites for venom toxins, offering a transformative approach to antivenom design. The project leverages state-of-the-art protein engineering, expression systems and mass spectrometry, coupled with comprehensive in vitro and in vivo venom neutralization assays, to outline methods for next-generation antivenoms that prioritize efficacy, safety and accessibility. To further refine venom profiling and assess antivenom efficacy, the ADDovenom project incorporates multi-enzymatic limited digestion to characterize toxin profiles from nine venomous snake species, including five *Dendroaspis* and four *Echis* species. Multi-enzymatic limited digestion improved peptide sequencing and identification, which surpass traditional trypsin workflows and enhance toxin characterization accuracy. In addition, a cutting-edge antivenomics technique combines magnetic beads with LC-MS and shotgun proteomics to streamline antivenom efficacy assessment. Applied to venom from the medically significant African viper *E. ocellatus*, this approach effectively identifies toxin-binding selectivity. Preliminary findings highlight the methodology effectiveness in rapidly characterizing antivenom efficacy. This innovative approach shows significant promise for advancing the development of more effective, targeted and economically accessible antivenoms, addressing the urgent needs posed by snakebite envenoming, especially in resource-limited regions.

**Keywords:** antivenom; mass spectrometry; venom

### 6.16. Neutrophil Activation by a Metalloproteinase Isolated from Bothrops jararacussu Snake Venom


**Milena Daniela Souza Silva ^1^, Vanessa Ferreira De Araujo ^1^, Carolina Pereira Da Silva ^1^, Erika Christina Santos De Araujo ^1^, Lívia Maria Vieira Brilhante ^1^, Mauro Valentino Paloschi ^1^, Alex Augusto Ferreira E. Ferreira ^1^, Micaela de Melo Cordeiro Eulálio ^1^, Hallison Mota Santana ^1^, Andreimar M. Soares ^2^, Sulamita da Silva Setubal ^1^, Juliana Pavan Zuliani ^1,3,^***


^1^ Laboratório de Imunologia Celular Aplicada à Saúde, Fundação Oswaldo Cruz, FIOCRUZ Rondônia, Porto Velho-RO, Brazil.^2^ Laboratório de Biotecnologia de Proteínas e Compostos Bioativos (LABIOPROT), Fundação Oswaldo Cruz, FIOCRUZ Rondônia, Porto Velho-RO, Brazil.^3^ Departamento de Medicina, Universidade Federal de Rondônia, UNIR, Porto Velho-RO, Brazil.

***** Correspondence: juliana.zuliani@fiocruz.br

**Abstract:** Snake venom metalloproteinases are enzymatic proteins present in large quantities in snake venoms, exhibiting proteolytic, hemorrhagic and coagulant activities. A snake venom metalloproteinase of class P-I, named BjssuMP-II, was isolated from the venom of *Bothrops jararacussu*. Studies have shown that this snake venom metalloproteinase induces neutrophil activation by releasing cytokines [tumor necrosis factor (TNF)-α, interleukin (IL)-1β, IL-6 and IL-8] and dsDNA, suggesting neutrophil extracellular trap release. This study aimed to evaluate the role of BjssuMP-II in the NLR family pyrin domain containing 3 (NLRP3) inflammasome complex activation in human neutrophils. Gene expression analysis by RT-qPCR was performed for NLRP3 inflammasome complex components, along with transcription factors and pro-inflammatory cytokines. Protein expression was evaluated using the Western Blot method to measure the protein levels that form the NLRP3 complex, including the NLRP3 apoptosis-associated speck-like protein containing a caspase recruitment domain (ASC), Caspase-1 and the constituent β-actin protein. Immunofluorescence was conducted to visualize NLRP3 and ASC puncta formation. Additionally, an immuno-enzymatic assay was performed to evaluate the NLRP3 inflammasome activation product, the IL-1β cytokine, in samples untreated and treated with the inhibitor of the NLRP3 sensor component (MCC950). The data showed that BjssuMP-II stimulates the gene expression of ASC, Caspase-1 and IL-1β after 30 min of incubation, while NLRP3, hypoxia-inducible factor 1-alpha (HIF-α), factor nuclear kappa B (NF-κB) and NIMA-related kinase-7 (NEK7) were observed after 1 h. Relevant protein expression was noted after 2 h for ASC and cleaved Caspase-1 proteins and for cleaved gasdermin D after 4 h. The release of mature IL-1β was detected after 18 h, and the MCC950 inhibitor reduced the release of IL-1β, confirming NLRP3 activation. Taken together, these findings indicate that BjssuMP-II induces the activation of the NLRP3 inflammasome complex.

**Support:** FAPERO, CNPq, CAPES

**Keywords:** metalloproteinase; neutrophil extracellular trap; NLRP3

## 7. Conclusions

The 30th Meeting on Toxinology of the SFET provided yet another excellent opportunity for productive discussions and exchanges on a variety of toxinology topics. It also highlighted the growing potential for further collaboration between national, European and international research groups in the field. Thanks to the generous support of the SFET, two prestigious awards—“Best Oral Presentation” and “Best Poster”—were given to two young researchers. The awardees were selected by a panel of experts after a lively and engaging discussion, which was made all the more challenging by the high quality of all the presentations.

We are confident that the recent advances in research on venoms and on toxins from a wide range of animal, plant, fungal, algal, mold and bacterial species will encourage the submission of numerous manuscripts to this special issue. We look forward to the continued growth and development of toxinology research and its many exciting applications.

Toxinology remains a highly versatile field, thanks to the extensive interactions between toxins and their living environments. These interactions are explored both in the context of unraveling the deep secrets of toxins in the pathologies they induce and in understanding their potential as models for the design of new diagnostic or therapeutic agents. The study of toxins offers valuable insights into disease mechanisms, while also opening up exciting possibilities for developing novel medical tools. The broad applications of toxinology highlight its importance in advancing both basic scientific knowledge and practical healthcare innovations.

## Figures and Tables

**Figure 1 toxins-17-00094-f001:**
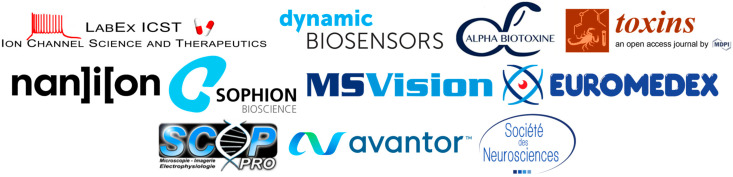
Sponsors’ logos.

**Figure 2 toxins-17-00094-f002:**
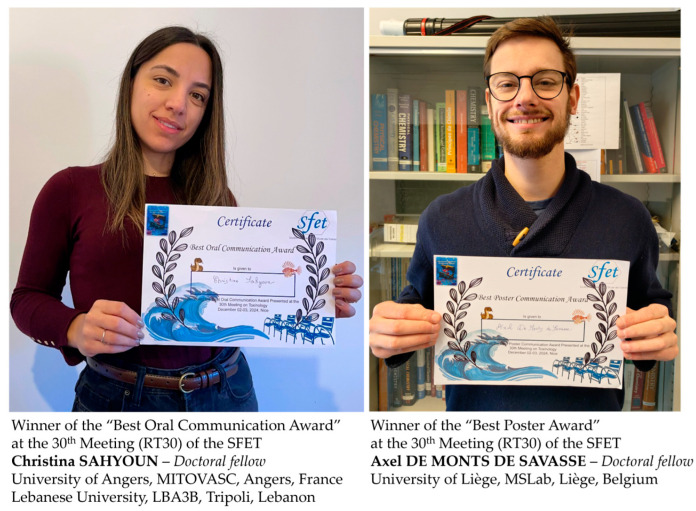
The “Best Oral Communication” and “Best Poster” Awardees at the 30th Meeting on Toxinology (RT30) of the French Society of Toxinology (SFET).

## Data Availability

No new data were created or analyzed in this study. Data sharing is not applicable to this article.

