# Peer review of "Report from the 30th Meeting on Toxinology, “Unlocking the Deep Secrets of Toxins”, Organized by the French Society of Toxinology on 2–3 December 2024"

_toxins, 2025, doi:10.3390/toxins17020094_

Round 1

Reviewer 1 Report

Comments and Suggestions for Authors

N/A

Author Response

We thank this reviewer for the fast and fully positive review of our ms.

Reviewer 2 Report

Comments and Suggestions for Authors

My suggestions for improving manuscript toxins-3464848 are the following:

1. I do not like some of the section titles. The title “1. Acknowledgements” is against current practices, when one would expect acknowledgements at the end; I would suggest to rename it “1. Premise and acknowledgements”, or something similar. The same applies to “2. Preface”, as it is not a preface but it describes the bulk of the meeting description, so it should be called something like “2. Meetings scope and topics” (or similar).

2. I would strongly suggest to mention the Special Issue as something ongoing in the present, rather than going to happen in the future.

- (page 3) “This issue will feature both the report from... [..]. We are confident that this Special Issue will appeal to...” might be changed to “This issue features both the report from... [..]. This special issue appeals to...”.

- (page 29) “Looking ahead, we are optimistic that the recent advances in venom and toxin research – from a wide range of animal, bacterial, fungal, and microorganism species – will encourage the submission of numerous manuscripts to this special issue” might be changed to “The recent advances in venom and toxin research – from a wide range of animal, bacterial, fungal, and microorganism species – encouraged the submission of manuscripts to this special issue”.

3. The images on page 7 should be either deleted or mentioned somewhere in the text. I would suggest to delete them, as they have no particular relevance.

4. Please refer to the Editorial Office of this journal for how to treat the name of the species in the titles. In fact, the name of the species should always be in Italics; however, once inserted in a title already in Italics, they were changed to plain text, and I am not sure this is the right choice. This issue affects titles such as 6.4, 6.13, and 6.16.

5. The Editorial Office should check whether the references included in the various sections (rather than in a classical single section “References”) are OK with the journal style.

6. Finally, the Editorial Office should be in charge also to check whether the section “Conflict of Interest” is OK, because reviewers cannot do it because the author information were removed for double blind peer review.

Other minor suggestions are:

- “These RT30 marked”. It should be either “This RT30” or “The RT30”.

- “Charlotte Odendall_*. Please check the underscore-like symbol before the asterisk.

- “Studies have shown that this SVMP induces neutrophil activation by releasing cytokines [tumor necrosis factor (TNF)-α, interleukin (IL)-1β, IL-6, and IL-8)”. The square bracket is closed with a round bracket.

- “between French, European, and international research groups in the field”. It might be more elegant to use “national” rather than “French”, or to omit it altogether.

Author Response

My suggestions for improving manuscript toxins-3464848 are the following:

We thank this reviewer for the detailed examination of our ms and useful suggestions.

1. I do not like some of the section titles. The title “1. Acknowledgements” is against current practices, when one would expect acknowledgements at the end; I would suggest to rename it “1. Premise and acknowledgements”, or something similar. The same applies to “2. Preface”, as it is not a preface but it describes the bulk of the meeting description, so it should be called something like “2. Meetings scope and topics” (or similar). Thanks for the suggestions - we modified the wording.

2. I would strongly suggest to mention the Special Issue as something ongoing in the present, rather than going to happen in the future. Agreed and not agreed, see below.

- (page 3) “This issue will feature both the report from... [..]. We are confident that this Special Issue will appeal to...” might be changed to “This issue features both the report from... [..]. This special issue appeals to...”.  Agreed and modified in part.

- (page 29) “Looking ahead, we are optimistic that the recent advances in venom and toxin research – from a wide range of animal, bacterial, fungal, and microorganism species – will encourage the submission of numerous manuscripts to this special issue” might be changed to “The recent advances in venom and toxin research – from a wide range of animal, bacterial, fungal, and microorganism species – encouraged the submission of manuscripts to this special issue”. Not agreed, because this conference report is aimed at being the first one (or one of the first) papers in the Special Issue. We modified as “We are confident that the recent advances in research on venoms and on toxins from a wide range of animal, plant, fungal, mould and bacterial species will encourage….”

3. The images on page 7 should be either deleted or mentioned somewhere in the text. I would suggest to delete them, as they have no particular relevance. Agreed and deleted.

4. Please refer to the Editorial Office of this journal for how to treat the name of the species in the titles. In fact, the name of the species should always be in Italics; however, once inserted in a title already in Italics, they were changed to plain text, and I am not sure this is the right choice. This issue affects titles such as 6.4, 6.13, and 6.16. As so per journal style, not per authors’ choice.

5. The Editorial Office should check whether the references included in the various sections (rather than in a classical single section “References”) are OK with the journal style. Checked by authors, OK now.

6. Finally, the Editorial Office should be in charge also to check whether the section “Conflict of Interest” is OK, because reviewers cannot do it because the author information were removed for double blind peer review. No comment from authors.

Other minor suggestions are:

- “These RT30 marked”. It should be either “This RT30” or “The RT30”. Agreed and modified.

- “Charlotte Odendall_*. Please check the underscore-like symbol before the asterisk. Deleted.

- “Studies have shown that this SVMP induces neutrophil activation by releasing cytokines [tumor necrosis factor (TNF)-α, interleukin (IL)-1β, IL-6, and IL-8)”. The square bracket is closed with a round bracket. Now closed with square bracket.

- “between French, European, and international research groups in the field”. It might be more elegant to use “national” rather than “French”, or to omit it altogether. Agreed and modified.

Reviewer 3 Report

Comments and Suggestions for Authors

Dear authors

The document is fine but needs some critical improvement.

1-      Kindly, differentiated between Toxinology as an institute or company name and toxicology as the name of the toxic science. Kindly, do that in the entire text.

2-      Kindly, follow the slandered criteria of the abbreviations.

3-      Kindly follow the slandered criteria of the style. I mean here to use wither US-English, or UK-English. Additionally, did not change from US-English, or UK-English and vice versa in the same abstract. If you have joined two words by ‘-‘, kindly do that in the entire abstract., etc.

4-      If you have described the source of the used toxins in a successive way (e.g. xxx toxin from animal and xxx toxins from xxx, and …) kindly describe the other toxins source as well to give a homogenize text.

5-      You might change the style by adding a number for each abstract. So, for example the reader can go through them by number as well.

6-      One could sense that the writer might not be a native English speaker, (me as well), but it might be better to let a native English speaker to wash the entire document.

7-      As a French languish learner, I could observe some French words which have the same meaning in English, but they are not the best to use in scientific documentation (e.g., jury).

8-      The image resolution need improvement. It might be better to use table to put the 2 images up and the text can be written down. In that case the text will be more readable.

9-      I have made some correction in the text, kindly revise.

10-   The abbreviation and other needed to be written. You might add them to the text.

a.       Table of Abbreviations

·          

·         APC        automated patch-clamp

·         ASIC       acid-sensing ion channels

·         CC           Climate change

·         CNRS     Centre National de la Recherche Scientifique

·         CT           computed tomography

·         DMTS    Département Médicaments et Technologies pour la Santé

·         GPCR     G-protein coupled receptors

·         HAB       Harmful algal blooms

·         HDP       Host DefenceDefense Peptides

·         HEK        human embryonic kidney

·         ICK         inhibitor cystine knot

·         ICST       Ion Channel Science and Therapeutics

·         LISM      Laboratoire d'Ingénierie des Systèmes Macromoléculaires

·         MELD    Multi-Enzymatic Limited Digestion

·         MSI        Mass Spectrometry Imaging

·         NAP       neurotoxin-associated proteins

·         NET        neutrophil extracellular trap

·         NGC       neurotoxin gene cluster

·         NSAF     Normalized Spectral Abundance Factor

·         NSCLC   Non-small cell lung cancer

·         NTNH    non-toxicnontoxic non-hemagglutinin

·         PET         positron emission tomography

·         PSG        posterior salivary gland

·         SBRC      Synthetic Biology Research Centre

·         SHFJ       Service hospitalier Frédéric Joliot

·          

·         These abbreviations are used but PerfectIt could not find a definition

·          

·         AHT

·         ASC

·         ATP

·         BoNT

·         CDT

·         CrLAAO

·         DMPS

·         EAC

·         ELISA

·         HepaRG

·         HPLC

·         IFN

·         IgG

·         LDH

·         MALDI

·         MDPI

·         MTT

·         MTX

·         MW

·         NADPH

·         NLR

·         PGN

·         PLOS

·         POPC

·         QT

·         RO

·         ROS

·         SA

·         SFET

·         SK

·         SVMP

·         TotB

·         UTCBS

·         WHO

Finally, I did not find a major correction(s), but the authors of this document need to get the chance to improve the document regarding to the above suggestions.

With my pleasure

Comments on the Quality of English Language

The English style need some improvements. Kindly read my comments.

1-      One could sense that the writer might not be a native English speaker, (me as well), but it might be better to let a native English speaker to wash the entire document.

2-      As a French languish learner, I could observe some French words which have the same meaning in English, but they are not the best to use in scientific documentation (e.g., jury).

Author Response

The document is fine but needs some critical improvement. Many thanks to this reviewer for the detailed examination of our ms and useful suggestions.

1- Kindly, differentiated between Toxinology as an institute or company name and toxicology as the name of the toxic science. Kindly, do that in the entire text. Not agreed - the word “toxinology” is related to peptidic or organic toxins of natural origin (animal venoms, plants, fungi, bacteria, etc) whereas “toxicology” is related to toxic compounds whatever their origin and chemical nature - and “Toxins” is the name of the journal…

2- Kindly, follow the slandered criteria of the abbreviations. Most abbreviations were already defined internally to each concerned abstract. However, we double-checked all of them and made substantial modifications, such as removing those rarely (or never) used and defining those that were not defined - hope we didn't miss critical ones.

3- Kindly follow the slandered criteria of the style. I mean here to use wither US-English, or UK-English. Additionally, did not change from US-English, or UK-English and vice versa in the same abstract. If you have joined two words by ‘-‘, kindly do that in the entire abstract., etc. We opted for British English for the entire ms and corrected hyphenation as thoroughly as possible.

4- If you have described the source of the used toxins in a successive way (e.g. xxx toxin from animal and xxx toxins from xxx, and …) kindly describe the other toxins source as well to give a homogenize text. Thanks for the suggestion - we specified “fungi or mould” for the mycotoxins.

5- You might change the style by adding a number for each abstract. So, for example the reader can go through them by number as well. We do not understand the suggestion - each abstract already has its own number.

6- One could sense that the writer might not be a native English speaker, (me as well), but it might be better to let a native English speaker to wash the entire document. The entire ms has been “washed” as thoroughly as possible (however, pls note that even native English (or French) speakers may not speak/write good English (or French)!).

7- As a French languish learner, I could observe some French words which have the same meaning in English, but they are not the best to use in scientific documentation (e.g., jury). “Jury” now reads “committee”, and we double-checked the other Gallicisms – hope we didn’t miss any.

8- The image resolution need improvement. It might be better to use table to put the 2 images up and the text can be written down. In that case the text will be more readable. We decided to delete the images (cf. item 3 of reviewer 2).

9- I have made some correction in the text, kindly revise. Many thanks for the detailed corrections and suggestions - we retained most of them.

10- The abbreviation and other needed to be written. You might add them to the text. Please confer to our reply to your item 2.

Finally, I did not find a major correction(s), but the authors of this document need to get the chance to improve the document regarding to the above suggestions. So we did, thanks again.

With my pleasure